# Learning the energy relaxation manifold from unrelaxed structures with RelaxNet

## Abstract

In an effort to bypass computationally expensive density functional theory (DFT) calculations for energy minimization and structure relaxation, rapid progress in the development of machine learning force fields/interatomic potentials (MLFF/MLIPs) and more robust models that adhere to quantum chemistry/physical paradigms and constraints have been realized. However, most research to date involves static-frame energy predictions only (i.e., given a specific atomic configuration, predict the energy of the current or final instance), neglecting intermediary physical insight-providing contexts. In this work, we developed RelaxNet, the first end-to-end, dynamics-aware, equivariant model that leverages neural ordinary differential equations (ODEs) and message passing neural networks (MPNNs) for predicting the energy relaxation landscape between the initial unrelaxed structure and final relaxed structure. From just the initial structure, which is often the configuration that is fed into DFT simulations, we can accurately recover the energy, forces, and geometric pathways for the trajectory at a competitive prediction accuracy, as evidenced by comprehensive benchmarking with state-of-the-art static models and MLIP-based relaxation methods. Additionally, we provide extensive insights on the use of implicit vs. explicit latent embedding evolution to offer perspectives on optimal strategies for future works that seek to integrate expensive graph-based neural networks and neural ODEs.

## 1 Introduction

Density functional theory (DFT) simulations are long-standing prerequisites for most conventional atomistic simulation methods (e.g., Grand Canonical Monte Carlo, molecular dynamics) that are governed by force fields. DFT can effectively optimize an unrelaxed structure to a more stable configuration by minimizing the potential energy landscape as guided by the forces acting on each atom in the structure. However, DFT is computationally expensive and often exhibit slow convergence, especially for large/complex systems. There is also a growing need for adaptable and scalable force fields that can easily extend to different molecular structure groups (e.g., metal-based compounds vs. inorganic molecules). To bypass these time-consuming energy-based methods and address these point, machine learning force fields (MLFFs) and interatomic potentials (MLIPs), which can be computed using surrogate models, have been proposed as viable and efficient methods of parameterizing atomic interactions with DFT-level accuracy (Botu et al.; Chmiela et al. (a;c); Unke et al.). These models generally emphasize representation learning, specifically developing adequate architectures and features that can capture the underlying latent physics and quantum mechanics principles that are intrinsic to these molecular systems. While there are many advances in this direction, there is currently no work devoted to expanding these concepts towards dynamics-level frameworks. Traditional one-to-one, static structure–property mappings can be useful for obtaining energies/forces at a discrete (single frame) state, but ultimately, this limits the scope of understanding by which a structure evolves towards equilibrium due to little reliance on transient fields. In short, instantaneous modeling neglects the self-consistency that is important in physics-abiding energy minimization processes.

The ability to model the DFT relaxation trajectory from a structure's initial unrelaxed state to its final relaxed state, however, can be valuable for learning smooth, continuous, and physical potential energy surface (PES) representations. To model these dynamics, an architecture that can inherently learn the derivatives and vector fields of the system, like neural ordinary differential equations

(ODEs), would be ideal. Neural ODEs, however, can be time-consuming and memory-intensive, especially when large networks, like graph-based models, are used as the ODE backbone, since this method requires numerical integration evaluation at each solver step. The intermediate states are usually stored as well, which could similarly lead to memory overflow. Hence, we need to either (1) develop an efficient, yet sufficiently expressive, neural representation that can be updated in each ODE step, or (2) reformulate how the latent dynamics are updated within the model. In this work, we explore both options in depth to inform appropriate model selection in the future. We bridge these primary knowledge gaps by introducing RelaxNet, a physics-informed, fully equivariant, neural ODE-based model that can learn energies and force fields at the intermediate and final states from only the initial state. This allows for defined dynamics at every point in space, smooth gradients in between frames, and physically-consistent continuous updates, which encourages better generalizability and robustness to noise/perturbations. RelaxNet can also be used to aid existing DFT efforts in parallel by predicting near-equilibrium states that are more favorable starting points for DFT simulations, thus reducing the expected convergence time (as opposed to starting from a purely unrelaxed configuration). As the first end-to-end trajectory-learning work in the MLFF space, this framework can contribute greatly to advancing equivariant ODE-based models in addition to their applications in scientific domains.

To summarize, this work makes the following key contributions:

1. We propose ***RelaxNet, a new and robust fully-equivariant neural ODE model*** that is capable of learning smooth surrogate dynamics exhibited in DFT relaxation. This is the ***first end-to-end, physics-informed modeling framework*** of its kind for predicting the relaxation trajectory based on energy and dynamics-conserving principles.

2. Provided only the initial unrelaxed atomic configuration, the model can predict the energies and forces for $n$ frames of the trajectory at the competitive accuracies demonstrated by other state-of-the-art models, as substantiated by benchmarking tasks with static models and post-training, MLIP-driven structure optimization methods.

3. We analyzed the effects of using implicit vs. explicit latent embedding evolution in neural ODEs to inform future relaxation trajectory prediction works that marry equivariant graph-based and ODE-based neural networks.

## 2 RELATED WORK

**Equivariant graph neural networks.** Since a crystal structure can be readily represented by an undirected graph, graph neural networks (GNNs) are appropriate frameworks for encoding local atomic and spatial information and forming high-quality molecular structure representations by means of message passing. Fundamentally, a graph, $G = (U, E)$, can be deconstructed into nodes, $u_i \in U$, and edges, $e_{ij} \in E$. To uniquely define each component, we can further impose node- ($h$ - e.g., electronegativity, atomic number) and edge-based ($r$ - e.g., bond distance) features. A general graph-based message passing workflow is outlined in Eqs. (1a) and (1b), where $m$ is the message used to update the nodal representation, $\sigma_a$ is an aggregative operation, $\phi_m$ is the message passing function, and $\phi_u$ is a generic update function.

$$\mathbf{m}_{ij} = \sigma_a\big(\phi_m(\mathbf{h}_i^{(l)}, \mathbf{h}_j^{(l)}, r_{ij})\big), \forall j \in \mathcal{N}(i) \tag{1a}$$

$$\mathbf{h}_i^{(l+1)} = \phi_u\big(\mathbf{h}_i^{(l)}, \mathbf{m}_{ij}\big) \tag{1b}$$

However, for anisotropic, vectorial representations, directionality becomes important, because the physical effects depends not only on the scalar magnitude, but also on the orientation relative to other atoms or reference frames. To ensure physical consistency, scalar properties ($S$) that are independent of the coordinate frame should be invariant to translations, rotations (defined by a rotation matrix, $\mathbf{Q}$), and permutations (Eq. (2a)). Vectorized properties ($\mathbf{V}$), on the other hand, should be equivariant to rotations, such that rotating the coordinate system (defined as $\mathbf{R}$) by $\mathbf{Q}$ should also rotate all vectors by $\mathbf{Q}$ (Eq. (2b)). Since vectors depends only on relative positions, translational invariance is naturally satisfied. In essence, equivariant models should be employed to ensure that these behaviors are respected.

$$S(\mathbf{R}) = S(\mathbf{R} + \Delta\mathbf{x}) \text{ (translation)} = S(\mathbf{R}\mathbf{Q}) \text{ (rotation)}, \quad \mathbf{Q} \in SO(3) \tag{2a}$$

$$\mathbf{V}(\mathbf{R}\mathbf{Q}) = \mathbf{Q}(\mathbf{V}(\mathbf{R})) \tag{2b}$$

Several methods of establishing equivariance in graph-based networks have been proposed. NequIP (Batzner et al.), for example, uses E(3)-equivariant convolutions to describe geometric tensor interactions. These convolution filters are based on learnable radial functions and spherical harmonics (to achieve rotational invariance). Other works have extended equivariance to an E($n$)-space, as proposed by Satorras et al. (EGNN) and Mao et al. (ENINet), which ensures invariance with respect to translational, rotational, and reflective operations. In the former work, spherical harmonics were not used; instead, equivariant graph convolution layers were established with only distance-based inputs. In the latter study, equivariant vectorial representations were introduced using directional message passing *via* many-body tensor representation (MBTR).

**Energy and force field prediction.** Naturally, these equivariant models can be adapted for energy (scalar) and force field (vector) prediction, which is particularly relevant for molecular modeling and DFT. In one such case, symmetric gradient domain machine learning (sGDML) (Chmiela et al. (b)) model is used to reproduce global PES for molecules with a few dozen atoms. Other works like DeepEF (Wu et al.) and DeePMD (Wang et al.) predict energy/forces *via* an atomic self-attentive model coupled with a geometric optimizer (in the former case) and deep, rotationally-invariant neural network (in the latter case). Researchers have also explored the use of genetic/evolutionary algorithms (Bin Faheem et al.) and KNN/random forest regressors (Nakata & Bai) for obtaining optimized force field parameters. In recent years, many studies have developed more robust ML frameworks for modeling quantum interactions and predicting energy (e.g., internal energy, atomization energy, in addition to other thermodynamic properties (e.g., enthalpy, Gibbs free energy, using the QM9 database, which features small, organic molecules. SchNet (Schütt et al.), for example, introduced continuous-filter convolutional neural networks that more accurately models local correlations, that allows for non-discretized PES reconstruction. In 2020, DimeNet (Gasteiger et al. (b)) (and later, DimeNet++ (Gasteiger et al. (a))) established directional message passing, which embeds messages passed rather than the atoms themselves. Later in 2023, Allegro (Musaelian et al.) presented a locally-equivariant deep neural network that doesn't use atom-centered message passing. Chen & Ong also developed M3GNET, a universal interatomic potential that incorporates three-body effects, unlike earlier models (e.g., CGCNN) that only consider pairwise interactions. Finally, there is ALIGNN (Choudhary & DeCost), a GNN-based model that performs message passing on the interatomic bond graph and line graph corresponding to bond angles.

**Molecular structure relaxation.** Earlier works on structure relaxation have also leveraged machine learning methods to obtain or accelerate the search for stable crystal configurations, although this area is currently not as well-explored as general MLIP development. In one such study, an iterative active learning approach was used alongside machine learning interatomic potentials (e.g., moment tensor potentials (MTPs)) to construct interatomic interaction models, thereby facilitating rapid crystal structure prediction (Podryabinkin et al.). Another study combined finetuned-MLFF and machine-learning surrogate models for learning reconstruction motifs and optimizing initial sampling structures to achieve energy-minimized structures, which can assist in point defect predictions (Mosquera-Lois et al.). Similarly, the introduction of DeepRelax (Yang et al.), a non-iterative equivariant graph neural network (EGNN)-based generative model that can predict relaxation quantities, has enabled low-energy crystal determination. Aside from surrogate structure relaxation models, post-trained MLIPs are also used in conjunction with external optimization algorithms to relax unstable conformations. It should be noted that a well-trained MLIP is a prerequisite for this method; insufficient training can lead to divergence. Earlier works (Fu et al.) have also suggested that accurate prediction of forces may not always translate to good reconstitution of dynamic trajectories.

## 3 BACKGROUND & PRELIMINARIES

**Energy minimization and force field calculations.** For a given PES, which can be defined by the total potential energy of a system (that is a function of the atomic coordinates, $\mathbf{r} = \{r_1, r_2, ..., r_N\} \in \mathbb{R}^{3N}$), we can minimize the energy, $E = E(\mathbf{r})$, to achieve the most stable molecular configuration. To enforce energy conservation and directly couple energy and force, we define the force acting on each atom in the $x, y, z$-directions by taking the negative gradient of energy with respect to position, as shown in Eq. (3). By jointly considering energy, force, and atomic coordinates, we ensure physical consistency and allow the model to learn dynamic updates that mirror common iterative techniques, like Velocity Verlet and quasi-Newton algorithms.

$$\mathbf{F}(\mathbf{r}) = -\nabla_r E(\mathbf{r}) \;\Rightarrow\; \mathbf{F}_i(\mathbf{r}) = -\partial E(\mathbf{r})/\partial \mathbf{r}_i \text{ for } i = 1, ..., 3N \tag{3}$$

**Modeling smooth surrogate relaxation dynamics.** Neural ODEs can be used to capture the continuous, physics-based relaxation dynamics *via* hidden state evolution, as generally defined in Eqs. (4a) and (4b). The evolution of the latent state $\mathbf{s}(t)$ at pseudotime $t$ is determined by an ODE, in which the governing dynamics are parameterized by a neural network $f$. A visualization of the relaxation trajectory is shown in Figure 1.

$$\frac{d\mathbf{s}(t)}{dt} = f(\mathbf{s}(t), t; \theta_p) \tag{4a}$$

$$\mathbf{s}(T) = \mathbf{s}(0) + \int_0^T f(\mathbf{s}(t), t; \theta_p)dt \tag{4b}$$

For physical systems, however, the ODE system (shown below), in its simplest form, can be defined as a function of the position ($\mathbf{r}(t)$) and velocity ($\mathbf{u}(t)$), like in Newton's second law. In this case, the force ($\mathbf{F}$) is derived from energy, which is a learnable neural network.

$$\frac{d}{dt}\begin{bmatrix}\mathbf{r}(t) \\ \mathbf{u}(t)\end{bmatrix} = \begin{bmatrix}\mathbf{u}(t) \\ \frac{1}{m}\mathbf{F}(\mathbf{r}(t))\end{bmatrix}$$

It should also be noted that DFT time, in this context, is not real physical time, but rather a pseudotime (similar to those in score-based generative and normalizing flow models). Since we are observing the problem from an optimization standpoint, quasi-Newton methods with artificial timesteps should suffice as long as the natural order of the relaxation trajectory is preserved. The learnable network should intrinsically learn the surrogate progression rule (i.e., surrogate dynamic) to equilibrium, since force is still evaluated at each step.

Figure 1: A high-level visualization of the relaxation trajectory from the initial unrelaxed state to final relaxed state, as governed by forces (and structure-level energy minimization).

# 4 METHODS

## 4.1 ENFORCING EQUIVARIANCE

**Spherical harmonics.** To establish equivariance and maintain translational/rotational invariance, explicit Cartesian inputs should be avoided. Instead, we can map the Cartesian space onto a spherical surface. Spherical harmonic expansion, for example, can be used to enforce rotational invariance, since it is constituted by orthonormal integration over all angles. Spherical harmonics, $Y_m^{(l)}(\theta, \varphi)$, form basis functions for angular-dependent models (and follows Wigner D-matrices under rotation) by encoding the angular position of the node, and by extension, the global rotational symmetries. The spherical harmonics are given by Eq. (5), where $N_m^{(l)}$ is the normalization constant that enforces orthonormality, $\exp(im\varphi)$ is the azimuthal component, and $P_m^{(l)}(\cos\theta)$ are the Legendre polynomials. Here, $\theta$ is the polar angle/colatitude, defined as $\theta = \arccos(z/r)$, $\phi = \arctan(y/x)$, $l$ is the orbital angular momentum quantum number, and $m$ is the magnetic quantum number ($-l \leq m \leq l$). The quantum numbers ultimately dictate the complexity of the spherical harmonic embeddings.

$$Y_m^{(l)}(\theta, \varphi) = \sum_m N_m^{(l)} \exp(im\varphi) P_m^{(l)}(\cos\theta) \tag{5}$$

**Linking spherical Bessel functions and spherical harmonics expansion.** To incorporate local, relative spatial information (e.g., distance), we expand the spherical harmonics to include a radial coefficient, $c_m^{(l)}(r)$. By using relative displacement, translational invariance can also be preserved. This term can be expressed using (1) gaussian radial basis functions (RBFs), defined as $c_m^{(l)}(r) = \exp(-\beta(r - r_0)^2)$, (2) polynomial basis functions (that can also be used as an envelope function for radial functions to increase the general expressivity of the representation), and (3) spherical Bessel functions (defined in Eq. (6a) for Bessel function of the first kind), which can be used to capture wave-like properties as commonly seen in quantum mechanical systems (e.g., Schrödinger equation). RBFs alone are not as robust for molecular systems, since they only approximate, rather than fully capture, the oscillatory and decaying effects inherent to atomistic simulations. Additionally, RBFs do not abide by orthogonality, unlike spherical Bessel functions. Consequently, to

encapsulate the profile contributions that are unique to each formulation, both radial transformations are used in the edge encoding. The full spherical harmonics expansion, detailed in Eq. (6b), considers both angular and radial information.

$$c_m^{(l)}(r) = \sqrt{\pi/2r}\,J_{l+1/2}(r) \tag{6a}$$

$$f(r, \theta, \varphi) = \sum_{l=0}^{l_{max}} \sum_{m=-1}^{l} c_m^{(l)}(r) Y_m^{(l)}(\theta, \varphi) \tag{6b}$$

## 4.2 RELAXNET ARCHITECTURE

The primary inputs to RelaxNet are the invariant node features (Section 5.1) and the spherical harmonics-mapped edge features (Section 4.1). The overall network is depicted in Figure 2.

**Interaction layer.** The atomic interactions are encoded by interaction layers (DTNN module) that are comprised of a convolution-based GNN for capturing pairwise interactions and an MPNN (denoted as TripletMPNN) for higher-order, three-body interactions. The angles are computed for each three-atom system and are cosine-transformed and passed through a zonal spherical harmonics function (that neglects dihedral effects). These angular embeddings are concatenated with the bond distances and spherical harmonics-expanded radial features, and will serve as inputs to the two MLP blocks, $MLP_s$ and $MLP_v$, which will output scalar and vector messages, respectively. These scalar messages are responsible for updating the existing node features. The outputs of the $MLP_v$ block will be multiplied with an orthogonal basis set, $B$, that is constructed from the Gram-Schmidt algorithm, and are then aggregated on a per-node level. The outputs from the mixed vector embedding layers, $MLP_m$, are multiplied by the aggregated embeddings and added to the gated vector embeddings (calculated from $MLP_g$) to retrieve the updated vector features. Similarly, the positions/coordinates are updated by multiplying the outputs from the $MLP_c$ block with the direction unit vectors and subsequently adding them to the original position. Each DTNN layer embedding in the encoder block will be updated and concatenated to produce the final embedding.

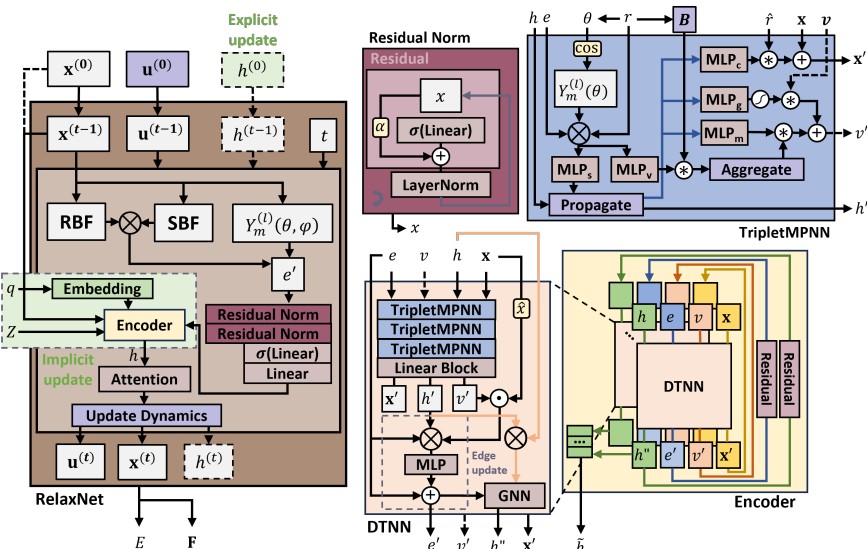

Figure 2: The **RelaxNet** architecture leverages a neural ODE wrapper to evolve the surrogate dynamics involved in DFT relaxation *via* either **implicit or explicit latent embedding update** steps. The latent embeddings, $h$, are computed from the **encoder block**, which is made up of **DTNN layers** with **residual** connections. The DTNN layer is comprised of **TripletMPNN layers** and a pairwise GNN. The dotted boxes/arrows indicate variables/blocks that can be excluded (vector messages, $v$) or interchanged ($h$ updates) in the overall model.

**Neural ODE.** We initialize the neural ODE with the states $\mathbf{x}^{(0)}$ and $\mathbf{u}^{(0)}$, where $\mathbf{x}^{(0)}$ is the initial unrelaxed position and $\mathbf{u}^{(0)}$ is a learned (initial velocity) embedding from the spherical harmonics-expanded edge features, for the initial structure. The encoder block produces interaction-based embeddings based on the position of the atoms at each pseudotime step. These representations are

then passed to an attention block. Finally, we feed these scalar embeddings to an MLP block to get the per-structure energy, after which the per-atom forces are computed by taking the negative gradient of energy with respect to atomic position. We then update the surrogate dynamics states by setting $d\mathbf{x}/dt = \mathbf{u}$ and $d\mathbf{u}/dt = \hat{\mathbf{F}}/m$, where $m$ is the atomic mass. For the ODE solver, we used a `rk4` fixed step solver with a relative tolerance of `rtol=1e-3` and an absolute tolerance of `atol=1e-3`.

**Implicit vs. explicit latent embedding evolution.** We experimented with two different methods of latent embedding updates. When the latent embeddings evolve implicitly (i.e., the interaction embedding is not an initialized state for the neural ODE), the latent variable, $h$, is recomputed for each evolved positional state, $\mathbf{x}^{(t)}$. On the contrary, the $h$ for explicit latent embedding evolution is established as an initial state (denoted as $h^{(0)}$), and is calculated using the same encoding block (but at the initial unrelaxed position) and updated *via* a MLP for each pseudotime step.

## 5 EXPERIMENTS

We performed several experiments to quantify the performance of RelaxNet. First, we establish a baseline model with only the encoder block (no neural ODE) to examine the effects of the number of RelaxNet base (DTNN) layers on energy prediction quality. Next, we explore the impacts of using implicit vs. explicit latent embedding evolution by comparing the energy/force mean absolute error (MAE) and training time for various cases by modulating the number of frames and structures. We then demonstrate successful recapitulation of meaningful physical properties, like energy, force field, and relaxation geometric pathway. To study model robustness, we further evaluate initial structure perturbation effects on overall prediction quality. Finally, we benchmarked RelaxNet with other state-of-the-art static energy prediction models and MLIP-based relaxation methods. The training configurations are detailed in Section A.1.

### 5.1 DATA

From the JARVIS database (containing over 80,000 materials), we selected the JARVIS DFT (3D) dataset for all experiments, due to its high conformational and compositional diversity (data distribution for different datasets can be found in Figure A.1). The dataset contains 33,473 unique crystal structures, 28,342 of which were used for training after featurization. From the `vasprun.xml` file of each structure, we extracted the atomic energy and per-atom forces (at different timesteps of the energy minimization process), the corresponding positions ($x, y, z$-coordinates), and the atomic numbers. We also obtained the total charges from the `OUTCAR` file. Finally, we retrieved the adjacency/connectivity matrix of each structure *via* crystallographic (`.cif`) files. The spatial and atom-level properties will define the input features, and the energy/forces will constitute the outputs.

### 5.2 BASE LAYER PERFORMANCE: STATIC, FINAL FRAME PREDICTION

We first consider a baseline model that is constructed from a single encoder block (no neural ODE envelope). For various training set sizes and different numbers of DTNN layers, we evaluate both the $MAE_E$ at the final frame and the average time it takes the model to achieve convergence (Table 1). The results indicate that using 4 DTNN layers yields higher $MAE_E$ and per-epoch training duration, which could be attributed to overfitting. Thus, for future experiments, we constrain our model to only 2 DTNN layers. The lower memory cost and faster runtime associated with using fewer layers would also be advantageous for regulating the neural ODE's computational efficiency. We similarly note that in using more unique training samples, the model attains better energy predictions.

To highlight the high prediction quality of the RelaxNet base layers, we further benchmarked our model with EGNN layers using the same sample splits, batch size (32), input features/embedding schemes, and number of layers (2 and 4). Although the EGNN layers are faster, RelaxNet base layers outperform the former in energy predictions across all training sizes. It is worth noting that good energy/force predictions are meaningful indicators of reliable MLIPs. Practically speaking, low-quality MLIPs often result in challenging or unstable convergence and can lead to poor relaxed conformation predictions. Accordingly, we trade computational speed for prediction accuracy, since RelaxNet's transient, state-dependent nature can allow errors to accumulate over time. To minimize these error propagation, high-quality predictions are imperative. Despite the longer per-epoch

training duration, we observe quick convergence trends, with 85% training convergence (based on validation set) occurring under ∼34 mins for all cases. The memory usage for RelaxNet base layers and EGNN are also comparable at 1584 and 1173 MB, respectively, for the 2-layer configuration.

Table 1: Global energy MAEs for RelaxNet base (DTNN) and EGNN model predictions at the final frame using 2/4 layers, computed for different number of samples. The 85%, 95%, and 99% convergence wall times are reported for the 2-layer configuration only.

| Model | $S$ | $MAE_E$ [eV] ($\downarrow$) | Time/epoch [min] ($\downarrow$) | Wall time [h:mm:ss] 85% | 95% | 99% |
|---|---|---|---|---|---|---|
| RelaxNet (DTNN) | 1,000 | **6.362**/16.947 | 1.24/2.10 | 0:22:17 | 0:35:09 | 0:51:58 |
| | 2,000 | **5.824**/13.103 | 2.32/3.74 | 0:05:26 | 0:12:25 | 0:45:09 |
| | 5,000 | **3.436**/8.606 | 5.39/9.38 | 0:33:49 | 2:00:50 | 7:21:45 |
| EGNN | 1,000 | 8.535/7.739 | **0.02**/0.02 | - | - | 0:01:56 |
| | 2,000 | 9.963/6.886 | **0.02**/0.02 | - | - | 0:02:27 |
| | 5,000 | 8.804/7.987 | **0.05**/0.08 | - | - | 0:26:16 |

## 5.3 ABLATION STUDY: TRAJECTORIAL LATENT STATE EVOLUTION

In the second experiment, we examine RelaxNet's performance by comparing the use of an implicit vs. explicit latent embedding evolution scheme. To also better understand the effects of trajectory length (i.e., the number of considered frames) and the number of unique crystal structures on prediction quality, we performed an auxiliary ablation study. Additionally, to ensure fair comparisons for each case, we collected all samples that have at least $n$ frames in the trajectory. We then sample $n$ equidistant frames from the full trajectory (including the initial and final frames).

In Table 2, we disentangle the effects of (1) the number of frames by using the same 5,000 structures (shaded rows) and (2) the number of total samples in the training set. For the explicit cases, the global $MAE_E$ and per-atom $MAE_F$ decreases with increasing frame count for the same 5,000 unique structures. This observation underscores the importance of intermediate frames, whereby each state conditions the evolution of future physical states, thus allowing for smoother relaxation trajectory reconstruction. We also note that after training with the same total number of samples, $S_{tot}$, the final and aggregated $MAE_E$ is comparable for both cases, while the $MAE_F$ is higher for the case with fewer unique samples but more frames, which highlights the importance of diversity and size of the training set. On the contrary, the implicit model results reveal that $MAE_E$ increases with number of frames. From this ablation study, we demonstrate that using explicit latent embedding updates is sufficient for obtaining full trajectory energy predictions with lower $MAE_E$ (17.4-65.1% better) and quicker per-epoch training time (63.0-90.6% faster) than the implicit scheme. This method would also be more suitable for large training sets.

Table 2: An ablation study comparing the implicit vs. explicit latent embedding evolution cases for different numbers of trajectory frames and unique crystal structures. The shaded rows indicate cases where the same structures are used. The global MAEs for energy/force predictions and the average training time per epoch are tabulated. The percent change from the baseline (2 DTNN layers) final frame $MAE_E$ and per-epoch runtime are reported. The percent change from the implicit all-frame $MAE_E$ and per-epoch runtime are also recorded.

| | $n$ | $S$ | $S_{tot}$ | Global $MAE_E$ (final/all) [eV] ($\downarrow$) | $MAE_F$ (final/all) [meV/Å] ($\downarrow$) | Avg. time per epoch [min] ($\downarrow$) |
|---|---|---|---|---|---|---|
| Implicit latent embedding evolution | 3 | 5,000 | 15,000 | 3.670/3.671 | 0.5/6.8 | 133.30 |
| | 5 | 5,000 | 25,000 | 5.053/5.042 (+47.1%) | 0.1/5.1 | 293.63 (+5347.7%) |
| | 10 | 5,000 | 50,000 | 6.457/6.472 | 0.3/4.4 | 311.25 |
| | 5 | 2,000 | 10,000 | 8.727/8.718 (+49.8%) | 0.8/8.1 | 61.88 (+2567.2%) |
| | 10 | 1,000 | 10,000 | 5.892/5.918 (-7.4%) | 0.5/12.0 | 31.16 (+2412.9%) |
| Explicit latent embedding evolution | 3 | 5,000 | 15,000 | 2.983/3.031 (-17.4%) | 0.8/3.1 | 17.95 (-86.5%) |
| | 5 | 5,000 | 25,000 | 2.764/2.781 (-44.8%) | 0.4/2.5 | 27.50 (-90.6%) |
| | 10 | 5,000 | 50,000 | 2.235/2.257 (-65.1%) | 0.3/2.3 | 49.84 (-84.0%) |
| | 5 | 2,000 | 10,000 | 4.582/4.652 (-46.6%) | 0.7/4.3 | 11.34 (-81.7%) |
| | 10 | 1,000 | 10,000 | 4.503/4.174 (-29.4%) | 1.5/10.8 | 11.54 (-63.0%) |
| | 5 | 24,311 | 121,555 | 2.919/2.996 | 0.0/0.5 | 136.80 |

## 5.4 Energy, Force, and Geometric Pathway Reconstruction

**Per-frame energy and force predictions.** We can further decompose the global energy and force errors into per-frame and cumulative MAEs, as depicted in Figure 3a, to elucidate the individual contributions of each state along the trajectory. In particular, we performed this analysis for the implicit latent embedding case with 10 frames/1,000 samples. From the MAE vs. frame index plots, it is evident that the per-frame energy and force MAEs decrease exponentially, with the lowest MAEs detected at the final relaxed frame, which is generally expected since the energy/forces decrease as the structure approaches a more stable configuration. Similarly, we observe that the cumulative probability curve shifts to the left for each relaxation step, indicating that there are lower errors with more relaxed frames. The overall energy predictions also show strong agreements with the ground truth energies (PCC=0.979, parity plot depicted in Figure A.2).

**Smooth structural relaxation trajectory.** In Figure 3b, we show that the inclusion of intermediate frames in the trajectory model (5-frame/5,000-structures case) improved the final-frame coordinate MAE, compared to directly using the initial frame to predict relaxed coordinates. This is indicated by the shifted distributions to lower MAEs for all frames, while discrete/static prediction produced an asymmetric, long-tailed distribution that parallels that of early frames (i.e., frame 0). This suggests that the instantaneously-predicted relaxed coordinates are largely shifted in a uniform and global manner, with little per-atom disentanglement. Conversely, with trajectory modeling, the distribution changes with frames due to more deliberate per-atom updates in ODE-based evolution, indicating refined atomic-level independence. This experiment exemplifies the importance of injecting transience into the modeling, since intermediate frames act as self-consistent physical constraints on the system. We also demonstrate that RelaxNet generates smooth relaxation trajectories, as reflected by the continuous and non-oscillatory geometric pathways in Figure 3c (shown for three random structures). These results highlight the advantage of learning derivatives, rather than absolute coordinates, to naturally model the gradual transitions inherent in the structure relaxation process. The coordinate offsets are measured by the pre-Kabsch MAE and post-Kabsch RMSD, where Kabsch refers to the alignment algorithm used to determine the optimal rotation matrix, such that the per-structure RMSD between two sets of points is minimized. The pre-Kabsch coordinate MAE and post-Kabsch RMSD are 0.0094 and 0.0241 $\mathring{A}$, respectively. Additional discussion on the energy-displacement relationship can be found in Section A.4. We also justify energy accuracy in our predicted final configurations based on the small final-frame force and geometry errors, and the inherent $\mathbf{F} = f(\mathbf{r}, E)$ parametric relationship. By the virial theorem and Taylor expansion of energy, energy deviations are also small by extension.

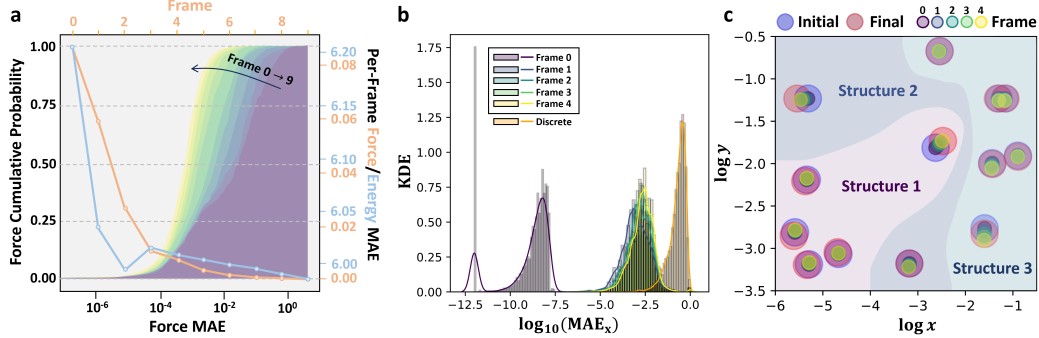

Figure 3: RelaxNet's (a) energy and force prediction performance for the implicit latent embedding evolution/10 frame/1,000 structure case, as quantified by the per-frame MAEs for both properties and the force cumulative density function. Additional geometric evaluations for the 5-frame/5,000 structure case were conducted, featuring a (b) histogram-based comparison of the per-frame (from the trajectory model) and the final-frame (from the static model) coordinate MAEs, and a plotted (c) smooth relaxation trajectory, from the initial to final state, for three random structures.

## 5.5 Effects of Conformational Perturbation on Prediction Quality

We assessed RelaxNet's robustness to noise by applying a perturbation factor ($\bar{\mathbf{x}}$) to the initial coordinates during both in- and post-training (results detailed in Table 3), such that $\bar{\mathbf{x}} = \text{scale} \cdot Z$, where $Z \sim \mathcal{N}(\mathbf{0}, \mathbf{I})$ and scale $\in \{\alpha, \beta\}$. The near-constant energy and force MAEs during post-training

perturbation (across all scales from $\beta = 0 \to 1$) indicate that the model has successfully learned the dynamics and is not blindly interpolating from the initial to final state. Perturbing the unrelaxed structures during training, however, slightly increased both energy and force MAEs, which suggests little benefit for generalizability.

| In-training perturbation scale ($\alpha$) | MAE | Post-training perturbation scale ($\beta$) | | | | |
|---|---|---|---|---|---|---|
| | $\beta =$ | $0 \to$ 1e-4 | 1e-3 | 1e-2 | 1e-1 | 1 |
| $\alpha = 0$ | E | 2.7594 | 2.7595 | 2.7596 | 2.7595 | 2.7553 |
| | F | 0.4 | 0.4 | 0.4 | 0.4 | 0.3 |
| $\alpha = $ 1e-2 | E | 4.8779 | 4.8779 | 4.8778 | 4.8776 | 4.8679 |
| | F | 0.8 | 0.8 | 0.8 | 0.8 | 0.8 |
| $\alpha = $ 1e-1 | E | 4.7677 | 4.7677 | 4.7677 | 4.7669 | 4.7675 |
| | F | 0.6 | 0.6 | 0.6 | 0.6 | 0.6 |

Table 3: The final-frame energy (E, eV) and force (F, meV/Å) MAEs for different combinations of in-/post-training perturbation configurations. The unperturbed state corresponds to the case in which $\alpha, \beta = 0$.

## 5.6 BENCHMARKING

**Assessing the performance of dynamic energy predictions relative to static approaches.** We start by comparing the RelaxNet model with explicit latent embedding evolution to other state-of-the-art energy prediction models (detailed in the JARVIS leaderboard from Choudhary et al.), like ALIGNN (Choudhary & DeCost), MatFormer (Cui et al.), PotNet (Lin et al.), and KGCNN (Reiser et al.), as shown in Table 4.

Specifically, we compared the $\mathrm{MAE}_E$ (derived from k-fold cross-validation on the test set) in our work for the 24,311 unique structure cases. The baselines are trained with all frames (i.e., includes intermediates), so we will use our all-frame $\mathrm{MAE}_E$ for proper comparison. Currently, the kgcnn_coGN can obtained the lowest $\mathrm{MAE}_E$ at 27.1 meV/atom, followed by kgcnn_coNGN at 29.1 meV/atom. We can also broadly deduce the models' performance with other datasets. For example, M3GNET (Chen

Table 4: Benchmarking RelaxNet with other static state-of-the-art energy prediction models. MAEs have units of meV/atom.

| Model | $S$ | $\mathrm{MAE}_E$ ($\downarrow$) |
|---|---|---|
| ALIGNN (all) | 55,713 | 33.1 |
| MatFormer (all) | 55,713 | 32.2 |
| potnet (all) | 55,713 | 29.3 |
| kgcnn_coNGN (all) | 55,713 | 29.1 |
| kgcnn_coGN (all) | 55,713 | 27.1 |
| RelaxNet (final/all) | 24,311 | 17.1/**24.6** |

& Ong), which is trained on the trajectory of 62,783 unique crystals (total of 187,687 samples) from the Materials Project (MP) database, yielded a $\mathrm{MAE}_E$ of 35 meV/atom. In the extensive benchmarking study conducted by Choudhary & DeCost that compared the $\mathrm{MAE}_E$ for different models across the MP and JARVIS databases, the authors noted that the models attained higher performance on the MP database, likely due to MP's larger dataset and lower target energies (50% lower) for MP structures. Overall, RelaxNet achieved an $\mathrm{MAE}_E$ of 17.1 (final frame) and 24.6 (all frames) meV/atom, which emphasizes the benefits of trajectory-based modeling, compared to instantaneous methods.

We want to emphasize, however, that this benchmarking study has a few qualifications. First, RelaxNet was trained with fewer samples overall since the neural ODE requires exactly $n$ frames, leading to a truncated database if $n > n_{\mathrm{tot}}$ ($n_{\mathrm{tot}}$ = total number of frames for the structure) or complete sample omission if $n < n_{\mathrm{tot}}$. As evident from Table 2, the $\mathrm{MAE}_E$ (especially on a per-atom level) generally decreases with more training samples. Second, the RelaxNet predicts the entire trajectory energies with a single initial frame (so intermediate energy predictions are also reliant on predicted intermediate states), while the other works predict the energies with absolute states. With these considerations and the benchmarking results in mind, we note that while this study does not provide a strict one-to-one comparison, we extrapolate that RelaxNet can achieve a relatively competitive prediction accuracy. We excluded force benchmarking because static models are trained on *all* frames (i.e., more intermediate frames with nonzero forces), whereas RelaxNet is trained on *only a subset* of intermediate frames, resulting in smaller $\mathrm{MAE}_F$. Since the energy magnitudes of the initial and final states are more comparable, this provides a fairer basis for comparison.

**Comparing RelaxNet with MLIP-based structure optimization.** In addition to the general improvements to energy-related predictions, we further extend our benchmarking to quantifying the predicted final frame coordinates' deviation from the ground-truth. After fully training the 2-layer EGNN model with all frames (of the 5000 unique structures), we relaxed the structure with the Atomic Simulation Environment (ASE) (Hjorth Larsen et al.) package, in which we implemented

a custom calculator that is purely governed by the trained MLIP. The structures are iteratively optimized with the LBFGS algorithm, with max step sizes of 0.2 $\mathring{A}$, a 0.05 eV/$\mathring{A}$ force-based stopping criterion (C1), and a maximum step threshold of 500 steps. We also explore a stopping criterion of 0.0751 eV/$\mathring{A}$ (C2) (RelaxNet final force norm). The RelaxNet and (EGNN) MLIP-derived final-frame energy/force deviations and post-training inference times are reported in Table 5 for an unseen test set of 250 structures. Evidently, we observe, for both force cutoff criteria, that RelaxNet (implicit and explicit models) produce energies and forces with significantly smaller MAE at notably lower inference time compared to those optimized *via* MLIP. While the EGNN achieved a test $MAE_E$ of ~18.26 eV after training, we noticed that during structure relaxation, the $MAE_E$ is higher during post-relaxation. The forces similarly indicate poorer convergence (for 4-layer case, see Section A.5). It is also worth noting that the post-training optimization inference scales with model depth, so careful tradeoffs are necessary to balance inference efficiency with MLIP accuracy. Performance-wise, this experiment highlights the strengths of a self-contained model. This improvement can also be rationalized by the fact that RelaxNet is trained end-to-end and guided by intermediate states, meaning it is always physically-constrained internally, so the predicted coordinates never diverge unrealistically from the initial state. Additionally, the model is fully coupled at the physical level (energy, force, and geometric changes), allowing for easy self-regulation in the event of discrepancies.

Table 5: A comparison of the final-frame energy MAE [eV], force MAE [eV/$\mathring{A}$], and inference times [m:ss] for an end-to-end RelaxNet and a post-training structure relaxation scheme (with stopping criteria C1/C2) using external ASE libraries/2-layer EGNN-based MLIP.

| Model | RelaxNet (explicit) | RelaxNet (implicit) | EGNN (C1/C2) |
|---|---|---|---|
| $MAE_E$ | 2.76 | 5.05 | 36.71/33.28 |
| $MAE_F$ | 0.0004 | 0.0001 | 0.64/0.76 |
| Inference time | 1:29 | 6:32 | 8:54/9:09 |

# 6 CONCLUSION

In this work, we developed RelaxNet, a robust, surrogate dynamics-aware neural ODE-based model with a fully equivariant backbone. We demonstrated the ability to evolve this descriptive, graph-based neural network ODE scaffold without memory explosion and within a reasonable training time. Moreover, unlike earlier works that predict energy/forces at static frames, for the first time, our model can predict the per-structure energy, per-atom forces, and conformations for a smooth and continuous DFT relaxation trajectory, provided only the initial unrelaxed coordinates. In addition, we thoroughly explored the effects of using implicit vs. explicit latent embedding evolution schemes *via* ablation studies. Afterwards, we benchmarked RelaxNet with other state-of-the-art static force field prediction models and MLIP-based structure optimization approaches, which revealed the exceptional predictive capabilities. The enhancements are conveyed through several means:

- Large improvements to final energy predictions compared to instantaneous modeling.
- Tight coordinate predictions evolving along a smooth geometric pathway, signifying model self-consistency that prevents unphysical moves that fall outside the desired distribution.
- Energy/force predictions that are robust to perturbations, revealing that the model learns the underlying dynamics rather than relying on a direct initial-to-final state interpolation.

To summarize, this work can expedite computationally-expensive DFT studies by (1) informing on the structure-level dynamics of the expected energy descent during relaxation (*via* metrics like energies/forces) and (2) providing users with more equilibrated (i.e., more stable crystal configuration) starting points for their DFT simulations.

# 7 LIMITATIONS & FUTURE WORKS

To our knowledge, we are the first to develop a dynamics-driven DFT relaxation trajectory prediction model with high-accuracy recapitulation of energies, forces, and geometric pathways from purely the initial unrelaxed state. Nevertheless, we recognize that the model could benefit from higher computational efficiency (i.e., faster training) with future iterations. Additionally, while the number of frames used in training RelaxNet has modest effects on prediction quality, there are opportunities to overcome the fixed-frame limitation by shifting the problem from a predictive to a generative framework. Transformers and video diffusion models that are conditioned on physics-based formulations can therefore be powerful alternatives, since they can generate each frame autoregressively, irrespective of trajectory length, and potentially increase generalizability.

ACKNOWLEDGMENTS

Hidden for anonymity.

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

# A APPENDIX

## A.1 TRAINING

**Loss function.** The model is directly supervised on several types of losses, including the per-structure (1) energy ($E$) and (2) energy landscape's monotonicity, and the per-atom (3) forces ($F$), (4) displacement ($\mathbf{x}$), and (5) directions. Here, $t$ is the frame number in a trajectory with a total of $T$ frames, $s$ is the sample number in a batch with $S$ structures, and $N$ is the number of atoms in the system. Loss terms 1, 3, and 4 simply ensure that the predicted values (e.g., $\hat{y}$) are close to their ground-truth values (e.g., $y$), while loss term 2 enforces a smooth energy function and a monotonically-decreasing energy throughout the relaxation trajectory. Mathematically, for each structure, a penalty is only applied if the predicted energy at the next step (frame t+1) is higher than the predicted energy at the previous step (frame t). This is enforced with the ReLU function. This ensures the energy decreases with each subsequent frame, because physically, the energy minimization process should see a monotonically decreasing trend. In the same vein, it enforces smoothness because it prevents non-monotonic jumps in the energy predictions, thereby eliminating potential oscillations. In addition to direct supervision with generic energy MAE loss, this term provides a meaningful physical constraint. Finally, loss term 5, modeled by cosine similarity, imposes a stricter alignment between the true and predicted directions. The loss function (Eq. (7)) is defined as the following, where $\lambda$ represents the weight for each loss term.

$$
\mathscr{L} = \frac{\lambda_1}{T \cdot S} \sum_{t=1}^{T} \sum_{s=1}^{S} \left| E_{t,s} - \hat{E}_{t,s} \right| + \frac{\lambda_2}{(T-1)S} \sum_{t=1}^{T-1} \sum_{s=1}^{S} \text{ReLU}\left( \hat{E}_{t+1,s} - \hat{E}_{t,s} \right)
$$

$$
+ \frac{\lambda_3}{T \cdot S \cdot 3N} \sum_{t=1}^{T} \sum_{s=1}^{S} \sum_{i=1}^{3N} \left( F_{t,s,i} - \hat{F}_{t,s,i} \right)^2 + \frac{\lambda_4}{T \cdot S \cdot 3N} \sum_{t=1}^{T} \sum_{s=1}^{S} \sum_{i=1}^{3N} \left( \mathbf{x}_{t,s,i} - \hat{\mathbf{x}}_{t,s,i} \right)^2
$$

$$
+ \lambda_5 \frac{\sum_{i \in \mathcal{M}} (1 - \cos \theta_i) \|\Delta \mathbf{x}_i\|}{\sum_{i \in \mathcal{M}} \|\Delta \mathbf{x}_i\| + \delta}, \quad \text{where } \mathbf{x} = \mathbf{x}_{t_f} - \mathbf{x}_{t_0} \text{ and } \cos \theta_i = \frac{\mathbf{x_i}}{\|\Delta \mathbf{x}_i\|} \cdot \frac{\hat{\mathbf{x}}_i}{\|\Delta \hat{\mathbf{x}}_i\|} \quad (7)
$$

**Hyperparameters.** During training, we use an AdamW optimizer with an initial learning rate of 5e-4 (ReduceLROnPlateau scheduler with a factor of 0.8 and an 8-epoch patience) and a weight decay of 5e-5. The model is trained over 1000 epochs with a training patience of 20 epochs to prevent overfitting. The train, validation, and test splits are 90/5/5. We used two NVIDIA Volta V100 GPUs (1 GPU = 32 GB), each accommodating a batch size of 64 crystals.

## A.2 DATASET SELECTION

We considered several databases for training RelaxNet. Among them are the JARVIS DFT (3D), rMD17, Materials Project (MP), and QM9 databases. Unlike the QM9 database, which has or-

ganic structures exhibiting only 5 unique elements and made up of less than 30 atoms, the JARVIS DFT database features structures with 86 distinct elements, including metals, and a maximum of 96 atoms. The rMD17 dataset contains MD trajectories for individual small organic molecules (e.g., aspirin, benzene, ethanol).

Since RelaxNet requires trajectories, only the JARVIS DFT and rMD17 datasets are appropriate candidates, since they both contain transient energy, force, and structural information. However, while the JARVIS DFT database features highly-diverse structures, both compositionally and conformationally, the rMD17 dataset has low compositional diversity and very small energy variances (-6306.4978 +/- 0.1048 eV for benzene). This limitation would severely hinder overall generalizability, and for this reason, we decided to proceed with the JARVIS DFT dataset, which exhibits a broad range of atomic elements, number of atoms, structural configurations, and energy magnitudes.

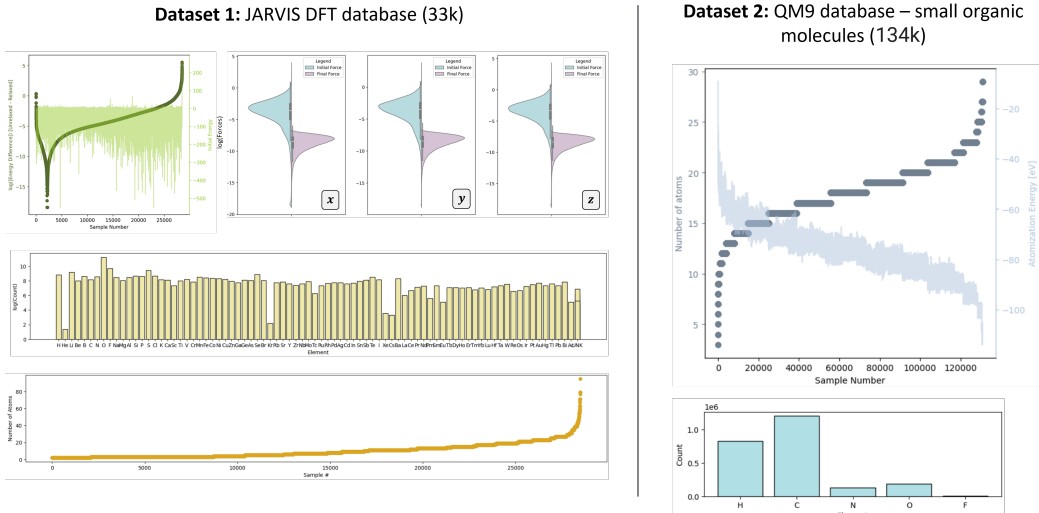

Figure A.1: The data distribution of the JARVIS DFT 3D database, featuring ∼33,000 different molecular structures, and the QM9 database, which consists of ∼134,000 small organic molecules. Only the initial and final structures' distributions are included in both datasets.

## A.3 ENERGY PARITY PLOT

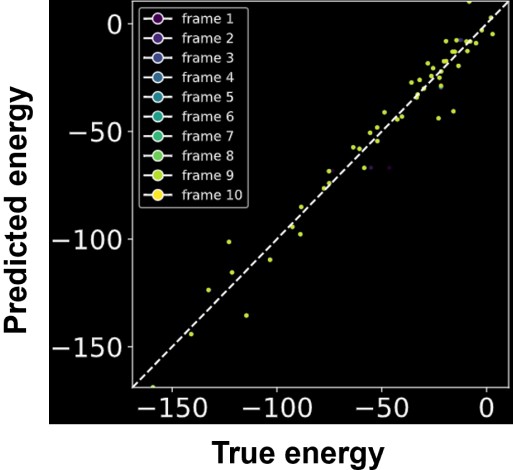

Figure A.2: A parity plot comparing the ground-truth and predicted energies (from the test set) for the implicit latent embedding evolution/10 frame/1,000 structure case.

## A.4 ENERGY-DISPLACEMENT RELATIONSHIP

We can also visualize the general effect of energy and average per-frame displacement $(\mathbf{x}_t - \mathbf{x}_0)$ on $\mathrm{MAE}_E$, as outlined in Figure A.3. First, we observed that most true displacements are concentrated between $10^{-3}$ and $10^{-2}$ Å. Second, the $\mathrm{MAE}_E$ decreases as the movement increases and energy magnitude decreases. This is consistent with earlier results, such that RelaxNet generated better final frame predictions, which could be attributed to the flatter energy landscape near equilibrium (e.g., smaller gradients), hence improving final energy predictions. Moreover, for higher energy magnitudes (regardless of signs), the resulting $\mathrm{MAE}_E$ is higher, as expected.

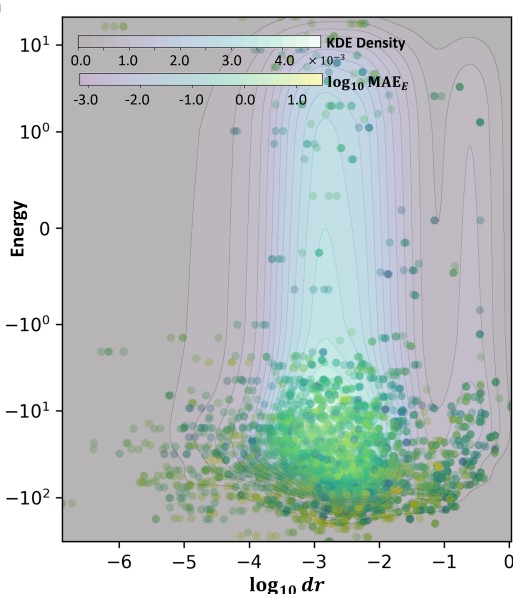

Figure A.3: The $\mathrm{MAE}_E$ (and overlayed KDE density) plotted against the energy and average per-frame displacement $(\mathbf{x}_t - \mathbf{x}_0)$.

## A.5 2-LAYER VS. 4-LAYER TRAINED EGNN COMPARISON FOR STRUCTURE RELAXATION

The EGNN was trained with a learning rate of 1e-4 and a training patience of 25 epochs (with a ceiling of 1000 epochs). We used the same dataset/splits, batch size, atomic number embedding scheme, and final energy head configurations as with the RelaxNet training. We noticed that the more the EGNN model is trained, the higher the forces become during structure relaxation. The inference time (during structure relaxation) is computed with the trained model set to "eval" mode when running the LBFGS algorithm.

Table 6: A comparison of the final-frame energy MAE [eV], force MAE [eV/Å], and inference times [m:ss] for a 2-layer trained EGNN and 4-layer trained EGNN with the structure optimization workflow applied afterwards.

| Model | 2-layer EGNN (C1/C2) | 4-layer EGNN (C1/C2) |
|---|---|---|
| $\mathrm{MAE}_E$ | 36.71/33.28 | 39.62/40.36 |
| $\mathrm{MAE}_F$ | 0.64/0.76 | 0.37/0.59 |
| Inference time | 8:54/9:09 | 25:29/25:09 |

