# OpenReview forum: "Learning the energy relaxation manifold from unrelaxed structures with RelaxNet"
_ICLR.cc/2026/Conference — Submitted to ICLR 2026_

### Official Review · Reviewer_J9gF · 2025-10-28

**Soundness:** 1
**Presentation:** 3
**Contribution:** 2
**Rating:** 2
**Confidence:** 3

**Summary:**

The authors tackle the task of predicting the per-structure energy and per-atom forces for a relaxation trajectory, given only the initial positions.

They propose to train a NeuralODE wrapped around a MPNN.
The ODE state consists of positions x(0) and learned “velocity” that is updated via the predicted force (obtained as gradient of the predicted energy).
The authors ablate two design choices to update the latent embeddings h, that are produced by the encoder block.
Implicit latent: h is not an ODE state. It’s recomputed from the current positions x(t) via the encoder at every pseudo-time step and then used to predict energy.

Explicit latent: h(0) is initialized and included as an ODE state variable, then updated each step by an MLP (along with the position and velocity)
They test their approach on crystal structure trajectories from the JARVIS database.

**Strengths:**

- Moves beyond single-frame energy prediction to trajectory learning between initial and relaxed states
- On JARVIS, authors report 17.06 meV/atom (final frame) vs. ~27–33 meV/atom for baselines, albeit with caveats

**Weaknesses:**

- It is unclear to me when one would be interested in predicting a relaxation trajectory compared to a MLIP that can do relaxation and any other dynamics (like MD)
- The approach limits data to sequential frames of fixed length
- The experiments only cover one not-so-common dataset with only 30k datapoints

**Questions:**

- I think the real benchmark is running relaxations with MLIPs. Can you compare the trajectory and final geometry error between relaxations with MLIPs, RelaxNet, and ground truth trajectories?
- I don’t quite understand how table 3 is computed. As I understand, RelaxNet will arrive at a different final structure than the data. Are the reported MAE_E for the baseline at the final RelaxNet-generated-structure or at the dataset-structure? For the ground truth, do you run DFT on the RelaxNet-generated-structure?
- You report only the MAE of the Energies in table 3, can you also include the Force MAE?
- It would be helpful to highlight (bold) the best numbers in the tables
- Can you measure the difference in inference time cost between running relaxations with an MLIP to using RelaxNet?
- What does this sentence in line 107 mean? "these equivariant models can be extended to periodicity-dependent applications, like molecular modeling and DFT”
- Does the use of a Neural ODE increase the cost of training compared to an MLIP by a lot?

---

> ### Author Response · Authors · 2025-11-23
> **Official Comment by Authors: Part 1**
>
> Thank you very much for putting the time into reviewing our manuscript and asking such thoughtful questions. The Reviewer’s insights were extremely valuable for helping us reframe our work into considering MLIP-based structure optimization methods, which greatly improved the comprehensiveness of and further solidified the motivations of our work.
>
> **Concerns about the requirements of fixed length.**
>
> We understand the concern that this could pose. While it would be attractive to produce longer trajectories if needed, a higher intermediate frame count is not a prerequisite for good prediction quality. As long as the initial and final frames are defined in the training set for each structure, RelaxNet will be able to predict the relaxation trajectory up to the final frame, regardless of whether 1, 5, or 8 intermediate frames are provided. Even with just 3 frames (initial + intermediate + final), the model should be able to arrive at the correct final relaxed state, with accurate energies, forces, and geometric configuration. The addition of intermediates, in this case, serves to guide the evolution of the relaxation trajectory by providing physical constraints that can bound the solution and prevent unphysical results. In the ablation studies we recorded in Table 2 (in the revised manuscript), specifically for the explicit latent state evolution case with the same number of unique structures, modulating the frame count still revealed low energy MAEs for all cases, meaning that RelaxNet is able to produce high-quality energy predictions despite different numbers of intermediate frame inputs. This also suggests that the model is generalizable to unseen test set structures of diverse composition and conformations. Please feel free to read our response to Reviewer w4Rk’s Q5, which also touches on this point.
>
> **Concerns about not-so-common dataset with only 30k datapoints.**
>
> Thank you very much for bringing up this great point. This would be a very valid concern for works that focus purely on static frame predictions, since there are many datasets suited for such training (e.g., QM9 for small organic molecules, Materials Project for solid state systems). However, publicly-available datasets containing DFT trajectories for a diverse set of structures are currently very limited. We also looked into the rMD17 dataset, which contains MD trajectories for individual small organic molecules (e.g., aspirin, benzene, ethanol) but observed very tiny energy variance (-6306.4978 +/- 0.1048 eV for benzene). While this dataset has conformation diversity, it contains little compositional diversity, which would hinder overall generalizability. For this reason, we decided to proceed with the JARVIS DFT dataset, which exhibits a broad range of atomic elements, number of atoms, structural configurations, and energy magnitudes. Based on our knowledge, we believe the JARVIS DFT dataset is most optimal for our tasks and has better potential for generalizability.

---

> ### Author Response · Authors · 2025-11-23
> **Official Comment by Authors: Part 2**
>
> **Q1. I think the real benchmark is running relaxations with MLIPs. Can you compare the trajectory and final geometry error between relaxations with MLIPs, RelaxNet, and ground truth trajectories?**
>
> **and**
>
> **Q5. Can you measure the difference in inference time cost between running relaxations with an MLIP to using RelaxNet?**
>
> **_Response_**: Thank you very much for the valuable insight and for suggesting this experiment. To emphasize the point on why RelaxNet could be a good alternative to post-trained MLIP-based structure relaxation, we added a new benchmark (Table 5) with a fully-trained EGNN-based relaxation, comparing the coordinate MAE, RMSD, and inference time. We also provide likely rationalizations as to why RelaxNet would perform better. The details can also be found in the revised manuscript as follows:
>
> >In addition to the general improvements to energy-related predictions, we further extend our benchmarking to quantifying the predicted final frame coordinates' deviation from the ground-truth. After fully-training the EGNN model (on the 5000-structure/2-layer case), we relaxed the structure with the Atomic Simulation Environment (ASE) package, in which we implemented a custom calculator that is purely governed by the trained MLIP. The structures are iteratively optimized with the LBFGS algorithm, with max step sizes of 0.2 A, a 0.05 eV/A force-based stopping criterion, and a maximum step threshold of 500 steps. The RelaxNet and (EGNN) MLIP-derived final-frame coordinate deviations and post-training inference times are reported in Table 5 for an unseen test set of 250 structures. The coordinate offsets are measured by the pre-Kabsch MAE and post-Kabsch RMSD, where Kabsch refers to the alignment algorithm used to determine the optimal rotation matrix, such that the per-structure RMSD between two sets of points are minimized. Evidently, we observe that RelaxNet output relaxed coordinates with significantly smaller MAE/RMSD at notably lower inference time compared to those optimized via MLIP. It is also worth noting that the post-training optimization inferencing scales with model depth, so careful tradeoffs are necessary to balance inferencing efficiency with MLIP accuracy. Performance-wise, this experiment highlights the strengths of a self-contained model. This improvement can also be rationalized by the fact that RelaxNet is trained end-to-end and guided by intermediate states, meaning it is always physically-constrained internally, so the predicted coordinates never diverge unrealistically from the initial state. Additionally, the model is fully coupled at the physical level (energy, force, and geometric changes), allowing for easy self-regulation in the event of discrepancies.
>
> **Q2. I don’t quite understand how table 3 is computed. As I understand, RelaxNet will arrive at a different final structure than the data. Are the reported MAE_E for the baseline at the final RelaxNet-generated-structure or at the dataset-structure? For the ground truth, do you run DFT on the RelaxNet-generated-structure?**
>
> **_Response_**: We apologize for the confusion. The ground truths are all derived from the DFT data reported in the JARVIS database. The RelaxNet model and baselines in the table are all trained on this dataset. The primary difference is the baselines are trained with more data and unique crystal structures, which we note as qualifications in the section. Since RelaxNet learns the trajectory from the initial unrelaxed to the final relaxed state, the model should ideally recapitulate the entire trajectory (from energy to geometry), including the final state. In this table (Table 4 in the revised manuscript), we retrieved the energy MAE for both the (1) final state only and (2) all frames in the trajectory (intermediate states + final state) across all structures. However, for evaluation, we compare the all-frame energy MAE only, since it is more reflective of what the other static models are trained on (intermediate + final frames). We will bold the number corresponding to the all-frame energy MAE to make this clear. We realized that we never mentioned this in the text, so we also added another line to offer more clarification.
>
> >The baselines are trained with all frames (i.e., includes intermediates), so we will use our all-frame $MAE_E$ for proper comparison.

---

> ### Author Response · Authors · 2025-11-23
> **Official Comment by Authors: Part 3**
>
> **Q3. You report only the MAE of the Energies in table 3, can you also include the Force MAE?**
>
> **_Response_**: Thank you for this astute observation. We excluded force benchmarking because static models are trained on all frames (i.e., more intermediate frames with nonzero forces), whereas RelaxNet is trained on only a subset of intermediate frames. This would naturally result in smaller force MAEs for RelaxNet, which we do not believe is a fair assessment of the other static models. Because the energy magnitudes of the initial and final states are more comparable, and there is generally a higher variance in the overall energy distribution (across all structures), we believe that this metric provides a fairer basis for comparison. We have included this note in the last few lines of the “Assessing the performance of dynamic energy predictions relative to static approaches” subsection (under the “Benchmarking” section) to address this exclusion.
>
> **Q4. It would be helpful to highlight (bold) the best numbers in the tables**
>
> **_Response_**: Thank you very much for this suggestion. We bolded the best numbers in Table 1 and Table 4 in the revised manuscript because there are clear comparisons made in these tables. We did not bold numbers in Table 2 because we noted improvements with percent changes.
>
> **Q6. What does this sentence in line 107 mean? "these equivariant models can be extended to periodicity-dependent applications, like molecular modeling and DFT”**
>
> **_Response_**: We apologize for the confusing language. We reworked the preceding subsection (“Equivariant graph neural networks”) to allow for a more generalized formulation of scalar invariance and vectorial equivariance, so we also rephrased Line 107 to the following to allow for a more natural segue to the next subsection (“Energy and force field prediction”).
> >Naturally, these equivariant models can be adapted for energy (scalar) and force field (vector) prediction, which is particularly relevant for molecular modeling and DFT.

---

> ### Author Response · Authors · 2025-11-23
> **Official Comment by Authors: Part 4**
>
> **Q7. Does the use of a Neural ODE increase the cost of training compared to an MLIP by a lot?**
>
> **_Response_**: Using neural ODEs does increase the training cost/time, but convergence is also generally quick (20-50 epochs). Nevertheless, we would like to emphasize several key points that can also **address the Reviewer’s general concern about the motivation of relaxation trajectory prediction**, particularly, the distinction as to when one should use RelaxNet and when MLIP-based optimization should be used.
>
> **_MLIP-based relaxation methods span across two different models_** (training MLIP and the subsequent relaxation steps). MLIPs are only trained end-to-end in energy/force, with structure optimization occurring afterwards, while RelaxNet leverages end-to-end joint energy/force/structure training across structures, allowing for a more unified, integrated, and physically-constrained architecture that leads to higher-fidelity optimization overall. Despite training on static intermediate frames, in its core, MLIPs only learn local physics, whereas RelaxNet learns the relaxation dynamics, which is more preferable in the grand scheme of trajectorial processes. This is apparent when comparing MLIP-based relaxation to RelaxNet. We observed that RelaxNet’s relaxed structure predictions will not diverge unreasonably from the original coordinate distribution and the initial unrelaxed state. However, with MLIP-based relaxation, this is not always the case, and the quality of the MLIP matters greatly (i.e., if the learned PES is poor, the convergence path towards a relaxed structure will also pose great challenge). Naturally, there is a clear tradeoff between training time, model depth (which also increases the post-training relaxation time), and final prediction accuracy (and similarly, convergence stability). From Table 5 (in the revised manuscript), we also observe that the optimization time is longer than RelaxNet’s inference time for 250 structures while also generating structures with higher RMSD when compared to the ground truth final conformation. We believe that architectural tuning between both training and post-training phases can also lead to time-intensive optimization. There is also a more general ease-of-use with an end-to-end, self-contained, self-regulating model like RelaxNet, which could make the utilization more attractive for researchers without extensive machine learning-specific domain knowledge.
>
> **_Once the model is fully-trained, the inference is quick and relies only on the initial unrelaxed state to reproduce the full relaxation trajectory._** For the inferencing stage, generalizability to unseen structures is important. Throughout our experiments, we demonstrated repeated generalizability:
>
> - Robustness to perturbation (newly-added Table 3 under the suggestion of Reviewer HMBn): very consistent energy and force predictions despite adding various magnitudes of noise to the initial unrelaxed structure, indicating that the model has learned dynamics and is not simply interpolating between initial and final structures
> - From Table 2, the explicit latent state evolution showed low energy MAEs across different numbers of frames for the same unique structure count, which suggests low frame count sensitivity during training and inferencing
>
> This is also additional justification for the use of the JARVIS DFT database, because it facilitates generalization to different structures due to the dataset encompassing a broad range of structures with different energy/force profiles, conformations, and compositions. For more specific structural groups, users can finetune (lower training time) with more targeted datasets.
>
> Thus, based on the large improvements to performance and easy, generalizable inferencing competency with capabilities of producing a smooth relaxation trajectory, we believe that the training time is an appropriate tradeoff.
>
> **_Additional motivation for using RelaxNet over MLIP-based structure relaxation_**
>
> Additionally, several works (e.g., Fu et al. [1]) have found that despite accurately predicting force fields, the reconstitution of dynamic trajectories may not always be as accurate. Such cases also motivate the need for an end-to-end model that can predict these optimization trajectories without instabilities or unphysically. We also made note of this in the “Molecular structure relaxation” subsection under “Related Works”.
>
> [1] Fu, X., Wu, Z., Wang, W., Xie, T., Keten, S., Gomez-Bombarelli, R. and Jaakkola, T., 2022. Forces are not enough: Benchmark and critical evaluation for machine learning force fields with molecular simulations. (https://openreview.net/forum?id=A8pqQipwkt)

---

> ### Author Response · Authors · 2025-11-23
> **Official Comment by Authors: Part 5**
>
> **A comment on the more general impact of our work**
>
> We would also like to emphasize that, at a higher level, we are providing a very general joint neural ODE and graph-based neural network framework, where we evaluate various methods for computational speedups. Since these integrated architectures are usually computationally-intensive (both in training time and memory efficiency), we were motivated to study the implicit vs. explicit latent state evolution schemes to provide some insights on this end. We hope our work can also be extended to other works and applications that deal with transient problems.
>
> We hope that our responses have addressed the Reviewer’s questions and that this discussion can alleviate concerns regarding the soundness of the work. We are happy to take any additional questions and comments, as we are always excited to improve our work!

---

> > ### Comment · Reviewer_J9gF · 2025-11-24
> >
> > Thank you for the clarifications!
> >
> > I have a few more questions:
> > 1. Thank you for adding relaxations with EGNN as a baseline. I think the RMSD to the ground truth minimum is a misleading metric here, as the minimum is not unique (especially under a learned energy surface). Can you report the final force norm of RelaxNet, then set the force-based stopping of LBFGS to the same number, and compare the wallclock of the MLIP to RelaxNet? Please also keep the existing results for the MLIP with a typical force-based stopping cutoff of 0.05 eV/A
> > 2. Can you report the force and energy MAE of the final frame? For relaxations the energy of the final frame is usually the interesting quantity for e.g. activation energies

---

> > > ### Author Response · Authors · 2025-11-26
> > > **Official Comment by Authors**
> > >
> > > Thank you again for these insightful questions, and in turn, helping us improve our work!
> > >
> > > **Q1. Thank you for adding relaxations with EGNN as a baseline. I think the RMSD to the ground truth minimum is a misleading metric here, as the minimum is not unique (especially under a learned energy surface). Can you report the final force norm of RelaxNet, then set the force-based stopping of LBFGS to the same number, and compare the wallclock of the MLIP to RelaxNet? Please also keep the existing results for the MLIP with a typical force-based stopping cutoff of 0.05 eV/A**
> > >
> > > **and**
> > >
> > > **Q2. Can you report the force and energy MAE of the final frame? For relaxations the energy of the final frame is usually the interesting quantity for e.g. activation energies**
> > >
> > > **_Response_**: This is a very good point. We completely agree with the Reviewer that the coordinate deviation-based metrics may be a bit misleading, considering the final configuration is not distinct/unique. We removed the pre-Kabsch coordinate MAE and post-Kabsch RMSD (and instead moved these values to the “Smooth structural relaxation trajectory” subsection under Section 5.4, since other reviewers were also interested in the positional reconstruction errors). As an alternative, we now report the $MAE_E$ and $MAE_F$, and kept the inference time, as the Reviewer suggested. Additionally, in response to the Reviewer’s first point, we’ve included all metrics for both stopping criteria, C1 (basic 0.05 eV/A cutoff) and C2 (RelaxNet final force norm). We’ve also created another column for RelaxNet (implicit) just to offer another mode of comparison. To fully address these points, we’ve also modified the main text to remove the coordinate-based error comparisons and added discussion on the energy/force MAE comparisons:
> > >
> > > > Evidently, we observe, for both force cutoff criteria, that RelaxNet (implicit and explicit models) produce energies and forces with significantly smaller MAE at notably lower inference time compared to those optimized \textit{via} MLIP. While the EGNN achieved a test MAE$_E$ of $\sim$18.26 eV after training, we noticed that during structure relaxation, the MAE$_E$ is higher during post-relaxation. The forces similarly indicate poorer convergence (for 4-layer case, see Section A.5).
> > >
> > > Additionally, we included a new section in the Appendix, entitled “2-layer vs. 4-layer Trained EGNN Comparison for Structure Relaxation” to compare the effects of using 2 and 4 layers for the EGNN training on the structure optimization afterwards. We also included supplementary EGNN training notes.
> > >
> > > We truly thank you again for the great suggestions, and please do not hesitate to reach out with more questions (or if additional clarifications should be made)!

---

### Official Review · Reviewer_HMBn · 2025-10-31

**Soundness:** 3
**Presentation:** 2
**Contribution:** 3
**Rating:** 4
**Confidence:** 3

**Summary:**

The paper introduces RelaxNet, a neural ODE-based model that predicts the entire DFT energy relaxation trajectory from just the initial unrelaxed structure.

**Strengths:**

This is a novel problem with a well-proposed method. Predicting the entire relaxation trajectory is both novel and of high practical value.

- provides good starting points for DFT simulations, reducing convergence time

- offers a clear comparison of implicit vs. explicit latent evolution across different trajectory lengths

- physics-based model

**Weaknesses:**

- Expensive training: 133-311 min/epoch for implicit method is prohibitive; even explicit (17-137 min) is slow. How scalable is this model
- No analysis of trajectory smoothness, physical plausibility, or whether intermediate states are actually useful
- limited dataset coverage

**Questions:**

- Beyond energy and force MAE, how well does RelaxNet reproduce the actual geometric pathway of the relaxation? For example, what is the average RMSD between the predicted and true final structures?

- The model is trained on DFT relaxation trajectories. If you initialized it with a perturbed structure that lies off the training trajectories (but within the distribution of atomic configurations), would it reliably find a path to a reasonable local minimum, or is it primarily learning to interpolate between seen initial and final states?

- How sensitive are the results to the choice of the ODE solver (RK4) and its hyperparameters (tolerances, step size)? Was any exploration done with adaptive solvers?

---

> ### Author Response · Authors · 2025-11-23
> **Official Comment by Authors: Part 1**
>
> Firstly, thank you very much for crafting this thorough review and providing valuable feedback on our work. The Reviewer’s thoughtful comments and concerns greatly helped us improve our work with suggestions on additional experiments and technical reasonings.
>
> **Regarding the concerns about expensive training and scalability.**
>
> This is a very valid concern. For our base layers, we justified the computational cost of the training (rationalization taken from the revised manuscript directly), as follows:
>
> >To highlight the high prediction quality of the RelaxNet base layers, we further benchmarked our model with EGNN layers using the same sample splits, batch size (32), input features/embedding schemes, and number of layers (2 and 4). Although the EGNN layers are faster, RelaxNet base layers outperform the former in energy predictions across all training sizes. It is worth noting that good energy/force predictions are meaningful indicators of reliable MLIPs. Practically speaking, low-quality MLIPs often result in challenging or unstable convergence, and can lead to poor relaxed conformation predictions. Accordingly, we trade computational speed for prediction accuracy, since RelaxNet’s transient, state-dependent nature can allow errors to accumulate over time. To minimize these error propagation, high-quality predictions are imperative. Despite the longer per-epoch training duration, we observe quick convergence trends, with 85% training convergence (based on validation set) occurring under ~34 mins for all cases. The memory usage for RelaxNet base layers and EGNN are also comparable at 1584 and 1173 MB, respectively, for the 2-layer configuration.
>
> Reviewer J9gF also suggested that we should compare RelaxNet to post-training MLIP-based structure optimization methods, and we fully agree with this feedback. However, earlier works (e.g., Fu et al. [1]) have suggested that accurate prediction of forces may not always translate to good reconstitution of dynamic trajectories. For this reason, an end-to-end model like RelaxNet, which has a more unified and physically-constrained approach to the problem, can be valuable for avoiding unphysical results. For this reason, we can justify the longer training time, especially since the prediction quality is high and results are physical.
>
> Regarding scalability, we were able to train 24,311 unique structures (5 frames each), making it a total of 121,555 samples. While the training time is long, convergence is quick, and the memory usage is low, so we do believe that the model is scalable. Additionally, inference is quick, provided that the model is trained, and we only need the initial unrelaxed configuration to generate the intermediate and final frames. For more specific structural groups, users can finetune the model (lower training time) on specific datasets. Nevertheless, even if we train on a smaller dataset, we have also demonstrated that our model is very generalizable, as indicated by (1) low energy, force, and final geometry MAEs for different number of sample sizes/frame counts and (2) high robustness to perturbation to the initial unrelaxed structure.
>
> Please feel free to also look over our response to Reviewer J9gF’s Q7, which addresses similar points.
>
> [1] Fu, X., Wu, Z., Wang, W., Xie, T., Keten, S., Gomez-Bombarelli, R. and Jaakkola, T., 2022. Forces are not enough: Benchmark and critical evaluation for machine learning force fields with molecular simulations. (https://openreview.net/forum?id=A8pqQipwkt)

---

> ### Author Response · Authors · 2025-11-23
> **Official Comment by Authors: Part 2**
>
> **About the limited dataset coverage.**
>
> Thank you very much for bringing up this point. This would be a very valid concern for works that focus purely on static frame predictions, since there are many datasets suited for such training (e.g., QM9 for small organic molecules, Materials Project for solid state systems). However, publicly-available datasets containing DFT trajectories for a diverse set of structures are currently very limited. We also looked into the rMD17 dataset, which contains MD trajectories for individual small organic molecules (e.g., aspirin, benzene, ethanol) but observed very tiny energy variance (-6306.4978 +/- 0.1048 eV for benzene). While this dataset has conformation diversity, it contains little compositional diversity, which would hinder overall generalizability. For this reason, we decided to proceed with the JARVIS DFT dataset, which exhibits a broad range of atomic elements, number of atoms, structural configurations, and energy magnitudes. Based on our knowledge, we believe the JARVIS DFT dataset is most optimal for our tasks and has better potential for generalizability.
>
>
> **Regarding the lack of analysis of trajectory smoothness, physical plausibility, or whether intermediate states are actually useful**
>
> **and**
>
> **Q1. Beyond energy and force MAE, how well does RelaxNet reproduce the actual geometric pathway of the relaxation? For example, what is the average RMSD between the predicted and true final structures?**
>
> Thank you very much for the helpful comment and question. From the Reviewer’s suggestion, we were able to include two new figures to support our response to the three points:
>
> **_Usefulness of intermediates states_**: we added a new Figure 3b (in the revised manuscript), where we
> >…show that the inclusion of intermediate frames in the trajectory model (5-frame/5,000-structures case) improved the final-frame coordinate MAE, compared to directly using the initial frame to predict relaxed coordinates. This is indicated by the shifted distributions to lower MAEs for all frames, while discrete/static prediction produced an asymmetric, long-tailed distribution that parallels that of early frames (i.e., frame 0). This suggests that the instantaneously-predicted relaxed coordinates are largely shifted in a uniform and global manner, with little per-atom disentanglement. Conversely, with trajectory modeling, the distribution changes with frames due to more deliberate per-atom updates in ODE-based evolution, indicating refined atomic-level independence. This experiment exemplifies the importance of injecting transience into the modeling, since intermediate frames act as self-consistent physical constraints on the system.
>
> **_Trajectory smoothness_**: we also added a new Figure 3c (in the revised manuscript), where we
> >…demonstrate that RelaxNet generates smooth relaxation trajectories, as reflected by the continuous and non-oscillatory geometric pathways (shown for three random structures). These results highlight the advantage of learning derivatives, rather than absolute coordinates, to naturally model the gradual transitions inherent in the structure relaxation process.
>
> **_Physical plausibility_**: we have demonstrated the ability to predict a smooth, continuous conformational trajectory along with their corresponding energy and force fields at competitive accuracies (as evidenced in the “Benchmarking” section). Additionally, we plot the geometric evolution from the initial to final state in Figure 3c to show that the model is capable of producing low-error relaxed coordinates. In the newly-added Table 5 (in the revised manuscript), we also computed the predicted and ground truth final coordinates, revealing a pre-Kabsch coordinate MAE of 0.0094 A and a post-Kabsch coordinate RMSD of 0.0241 A, which is significantly lower than those of the post-training (EGNN) MLIP-based optimized structure. Due to the physical constraints imposed by the model, in addition to the neural ODE formulation (learning derivatives enables the prediction of smooth gradients in between frames, and physically-consistent continuous updates) and guidance from the intermediate frames, the trajectory is smooth and will not deviate unrealistically from the original distribution.

---

> ### Author Response · Authors · 2025-11-23
> **Official Comment by Authors: Part 3**
>
> **Q2. The model is trained on DFT relaxation trajectories. If you initialized it with a perturbed structure that lies off the training trajectories (but within the distribution of atomic configurations), would it reliably find a path to a reasonable local minimum, or is it primarily learning to interpolate between seen initial and final states?**
>
> **_Response_**: This is an excellent suggestion. We added a new table (Table 3 in the revised manuscript) where we examine the effects of different perturbation factors on energy and force MAEs. We applied a perturbation to the initial unrelaxed structure (guided by the in-training perturbation scale (α), which we modulated from 0 to 1e-1) during the training process. We also studied the effects of applying a post-training perturbation to the initial structure during inferencing (guided by the post-training perturbation scale (β), which we modulated from 0 to 1). We observed very insignificant changes in the energy and force MAEs for all cases, indicating that RelaxNet is robust to perturbations and noise. This suggests that the model is not blindly interpolating between the initial and final states and is instead learning the underlying dynamics. To accompany the table, we also added a new section to the “Experiments” section, entitled “Effects of conformational perturbation on prediction quality” (Section 5.5), where we elaborate on our findings.
>
> **Q3. How sensitive are the results to the choice of the ODE solver (RK4) and its hyperparameters (tolerances, step size)? Was any exploration done with adaptive solvers?**
>
> **_Response_**: This is a great question. During the hyperparameter tuning phase, we empirically tested a fixed step solver (rk4) and an adaptive solver (dopri5), along with different learning rates (from 1e-5 to 1e-2) and tolerances (from 1e-5 to 1e-1). We observed that if the ODE becomes too stiff, the adaptive solver will encounter overflow errors, as expected. Similarly, exceedingly low absolute and relative tolerances can lead to training instabilities in the neural ODE. The fixed step solver, however, was very stable even through extended training intervals at all tested tolerances. However, we set a slightly higher tolerance (1e-3) to reduce the training time.
>
> We hope that our point-by-point responses to the questions and concerns have adequately addressed the issues raised. We are happy to continue the discussion should there be additional questions, comments, or feedback!

---

### Official Review · Reviewer_w4Rk · 2025-11-02

**Soundness:** 2
**Presentation:** 1
**Contribution:** 2
**Rating:** 2
**Confidence:** 4

**Summary:**

This paper introduces RelaxNet, a dynamics-aware and equivariant deep learning model for structure relaxation. To address the limitation of prior works that primarily predict energy for static structures while overlooking intermediate physical insights, RelaxNet leverages neural ordinary differential equations (neural ODEs) combined with message-passing neural networks (MPNNs) to model the energy relaxation landscape between initial and final structural states. Empirical results and extensive ablations are provided to support the rationale of the proposed approach.

**Strengths:**

* The idea of using neural ODEs to model relaxation trajectories is novel and thoughtfully implemented, offering a promising alternative formulation for this task.
* RelaxNet achieves competitive accuracy compared to state-of-the-art methods on standard benchmarks.

**Weaknesses:**

* While modeling intermediate states is intuitively appealing, it remains unclear why this should necessarily improve the prediction accuracy of the final relaxed state. As shown in Table 3, the performance gain over strong baselines is marginal. The authors should provide stronger intuition—or theoretical justification—for why this intermediate modeling leads to better final-state predictions.
* Insufficient experimental analysis:
  * There is no clearly defined, chemically meaningful accuracy threshold for evaluating relaxation performance, making it difficult to assess the practical utility of the reported results.
  * A comparison of training cost (e.g., wall-clock time, memory usage) between RelaxNet and baselines is missing. Given the modest performance improvement, such analysis is essential to evaluate the method's efficiency.
  * The requirement that training systems must contain more than $n$ frames (due to the neural ODE formulation) raises concerns about generalizability, since this limitation may restrict the applicability of RelaxNet to systems where long relaxation trajectories are unavailable.

**Questions:**

* In Equation (2b), the notation $G$ is used without a clear definition. Could the authors clarify its meaning?
* Could the authors provide more detailed explanations of the data processing steps mentioned in line 181?
* The second term in Equation (7), $\hat{E}_{t, s+1} - \hat{E}_{t,s}$, is difficult to interpret. Could the authors explain it more clearly?
* How exactly is the energy MAE computed for RelaxNet for evaluation?
* For systems with varying numbers of frames, the authors sample $n$ equidistant points. Does this result in different effective time intervals between samples? If so, could this introduce ambiguity in the learned dynamics or physical consistency of the gradient field?

**Details Of Ethics Concerns:**

No ethics concerns observed.

---

> ### Author Response · Authors · 2025-11-23
> **Official Comment by Authors: Part 1**
>
> We sincerely thank you for your review and detailed feedback on how we can improve our work. Through the Reviewer’s comments and concerns, we were able to find many areas in our manuscript to improve upon, from additional experiments to better presentation of our work.
>
> **Regarding the improvement of the final relaxed state predictions and the lack of justification for said improvements.**
>
> Thank you very much for this helpful remark. We agree with the Reviewer on this assessment and realized that we did not provide proper justification in the manuscript. We believe that RelaxNet improves both intermediate and final state predictions for several reasons. These rationalizations are also included in the revised manuscript.
>
> - First, **_neural ODEs naturally learn derivatives_**, rather than absolute values, so it models the gradual transitions inherent in the structure relaxation process. By learning in the intermediate states, rather than doing so instantaneously, it would allow for defined dynamics at every point in space, smooth gradients in between frames, and physically-consistent continuous updates, which encourages better generalizability and robustness to noise/perturbations. These smooth/continuous updates are especially important because each state conditions the evolution of future physical states, and thus, ideally, each step should have good predictions leading up to the final prediction
> - **_RelaxNet is trained end-to-end and guided by intermediate states_**, meaning it is always physically-constrained internally. Because the model is fully coupled at the physical level (energy, force, and geometric changes), this allows for easy self-regulation in the event of discrepancies during inferencing.
> - We also concretely see the smooth geometric evolution from initial to final state (Figure 3c in the revised manuscript). Naturally, improving the atomic position predictions of each state also enables more accurate energy/force/geometric property estimations between states, ultimately leading to better final energy predictions. To that same effect, while the improvements appear marginal, we believe that any improvement is important since the model is transient and heavily state-dependent (errors can propagate over time if intermediate representations are not adequate). We see significant improvement to the final energy as a result.
>
> **For the second concern about the lack of clearly defined, chemically meaningful accuracy threshold for evaluating relaxation performance and difficulty in assessing the practical utility of the reported results.**
>
> We completely agree with the Reviewer on this comment and have added new results to address this point. Some metrics that we believe can quantify the relaxation performance meaningfully are (1) energy, (2) force, and (3) structure MAE. First, the energy MAE can measure the accuracy of the potential energy surface (PES), which assesses the physicality of the energy minima and the molecular stability of a structure. Second, the force MAE can measure the error in which atom moves during relaxation. This will dictate the shape of the PES curvature and the path by which the atoms move towards equilibrium, thus capturing dynamics. Finally, the structure MAE can provide insights into several critical structural features, including bond length/angle and local and global atomic arrangement. To convey these metrics, we can use the newly-added Figure 3b-c to illustrate smooth, continuous predicted relaxation trajectories with the assistance of intermediate frames. Moreover, we can use Table 3-5  (in the revised manuscript) to show competitive energy, force, and geometry predictions.
>
> Regarding the practical utility of this work, RelaxNet can expedite computationally-expensive DFT studies by (1) informing on the structure-level dynamics of the expected energy descent during relaxation via metrics like energies/forces and (2) providing users with more equilibrated (i.e., more stable crystal configuration) starting points for their DFT simulations by using the predicted final state. Configurations that are more relaxed can decrease the DFT compute time and improve overall convergence. Additionally, this model provides users with an alternative to MLIP-based structure relaxation.

---

> ### Author Response · Authors · 2025-11-23
> **Official Comment by Authors: Part 2**
>
> **For the third concern about missing comparison of training cost (e.g., wall-clock time, memory usage) between RelaxNet and baselines.**
>
> Thank you very much for suggesting these new metrics. Because the only fair basis of comparison is to existing static models, we decided to update Table 1 with an additional benchmark comparing the RelaxNet base (DTNN) layer and EGNN layer. We included new time/epoch and wall time at 85%, 95%, and 99% saturation/convergence for the DTNN layers. The following details were also added to the revised manuscript to provide justification for the use of our layers rather than faster layers:
>
> >To highlight the high prediction quality of the RelaxNet base layers, we further benchmarked our model with EGNN layers using the same sample splits, batch size (32), input features/embedding schemes, and number of layers (2 and 4). Although the EGNN layers are faster, RelaxNet base layers outperform the former in energy predictions across all training sizes. It is worth noting that good energy/force predictions are meaningful indicators of reliable MLIPs. Practically speaking, low-quality MLIPs often result in challenging or unstable convergence, and can lead to poor relaxed conformation predictions. Accordingly, we trade computational speed for prediction accuracy, since RelaxNet’s transient, state-dependent nature can allow errors to accumulate over time. To minimize these error propagation, high-quality predictions are imperative. Despite the longer per-epoch training duration, we observe quick convergence trends, with 85% training convergence (based on validation set) occurring under ~34 mins for all cases. The memory usage for RelaxNet base layers and EGNN are also comparable at 1584 and 1173 MB, respectively, for the 2-layer configuration.
>
> As Reviewer J9gF also mentioned, a benchmark against a post-trained MLIP-based structure optimization method would be beneficial to the overall study. To further highlight why RelaxNet is a strong alternative, we added a new benchmark (Table 5) with a fully-trained (EGNN) MLIP-based structure relaxation that compares the MAE/RMSD and inference time for both methods. We also provide likely justification as to why RelaxNet would perform better. The details (found in the revised manuscript) are as follows:
>
> >In addition to the general improvements to energy-related predictions, we further extend our benchmarking to quantifying the predicted final frame coordinates' deviation from the ground-truth. After fully-training the EGNN model (on the 5000-structure/2-layer case), we relaxed the structure with the Atomic Simulation Environment (ASE) package, in which we implemented a custom calculator that is purely governed by the trained MLIP. The structures are iteratively optimized with the LBFGS algorithm, with max step sizes of 0.2 A, a 0.05 eV/A force-based stopping criterion, and a maximum step threshold of 500 steps. The RelaxNet and (EGNN) MLIP-derived final-frame coordinate deviations and post-training inference times are reported in Table 5 for an unseen test set of 250 structures. The coordinate offsets are measured by the pre-Kabsch MAE and post-Kabsch RMSD, where Kabsch refers to the alignment algorithm used to determine the optimal rotation matrix, such that the per-structure RMSD between two sets of points are minimized. Evidently, we observe that RelaxNet output relaxed coordinates with significantly smaller MAE/RMSD at notably lower inference time compared to those optimized via MLIP. It is also worth noting that the post-training optimization inferencing scales with model depth, so careful tradeoffs are necessary to balance inferencing efficiency with MLIP accuracy. Performance-wise, this experiment highlights the strengths of a self-contained model. This improvement can also be rationalized by the fact that RelaxNet is trained end-to-end and guided by intermediate states, meaning it is always physically-constrained internally, so the predicted coordinates never diverge unrealistically from the initial state. Additionally, the model is fully coupled at the physical level (energy, force, and geometric changes), allowing for easy self-regulation in the event of discrepancies.

---

> ### Author Response · Authors · 2025-11-23
> **Official Comment by Authors: Part 3**
>
> **Q1. In Equation (2b), the notation G is used without a clear definition. Could the authors clarify its meaning?**
>
> **_Response_**: Thank you very much for pointing this out. We realized that some variable naming conventions were inconsistent and that the “Equivariant graph neural networks” and “Energy and force field prediction” subsections were blurring into one another, so we decided to rework the two subsections to offer more clarity. In the revised manuscript, the “Equivariant graph neural networks” subsection will assume more generalized formulations, such that the transformation groups will act on scalar, *S*, and vectorial, **V**, properties, rather than energy or force explicitly. In addition, we renamed some variables, such that the coordinate system is now denoted as **R** and the rotation matrix as **Q**. Now, Equations 2a-b are as follows:
>
> *S*(**R**) = *S*(**R**+Δ**x**)  (translation) = *S*(**RQ**)  (rotation),        **Q** ∈ SO(3)    (2a)
>
> **V**(**QR**) = **Q**(**V**(**R**))                                                                                        (2b)
>
> To accommodate these changes, we also updated the corresponding paragraph with appropriate notations, additional details on scalar and vectorial transformations, and proper motivation for using equivariant models.
>
> **Q2. Could the authors provide more detailed explanations of the data processing steps mentioned in line 181?**
>
> **_Response_**: After some consideration, we decided to rephrase “post-processing” to “featurization”. The bulk of the featurization steps are detailed under the “Data” subsection, which mainly entails extracting the atomic information (e.g., element, total charges, position), per-structure global energy, and per-atom forces. From the crystallographic structure, we were able to extract the bond connectivity and adjacency matrices. We then build our dataset of multi-frame trajectories.
>
> **Q3. The second term in Equation (7), $\hat{E}{t, s+1} - \hat{E}{t,s}$, is difficult to interpret. Could the authors explain it more clearly?**
>
> **_Response_**: We apologize for the confusion. The second term in the loss function is used to enforce energy smoothness and monotonicity, but we mistakenly added and subtracted the 1s for the incorrect variable. The correct equation should be:
>
> $$
> L_2 = \frac{\lambda_2}{(T-1)S} \sum_{t=1}^{T-1} \sum_{s=1}^{S} \operatorname{ReLU} |\hat{E}(t+1, s) - \hat{E}(t,s)|
> $$
>
> Thank you very much for the great catch. This loss term has also been amended in the revised manuscript. Mathematically, for each structure, a penalty is only applied if the predicted energy at the next step (frame *t*+1) is higher than the predicted energy at the previous step (frame *t*). This is enforced by the ReLU function and ensures that the energy decreases with each subsequent frame, because physically, the energy minimization process should be monotonically-decreasing. In the same vein, it enforces smoothness because it prevents non-monotonic jumps in the energy predictions, thereby eliminating potential oscillations. In addition to direct supervision with generic energy MAE loss, this term provides a meaningful physical constraint.  We also moved the entire “Training” section to the Appendix (Section A.1) and have added this note in the “Loss” subsection to offer more clarification.
>
> **Q4. How exactly is the energy MAE computed for RelaxNet for evaluation?**
>
> **_Response_**: The model will predict an array of energies with shape R^(S×T), where S is the total number of structures in the batch and T is the number of frames in the trajectory. The energy MAE is then computed as shown in the following equation.
> $$MAE_E=\frac{1}{T\cdot S} \sum_{t=1}^{T} \sum_{s=1}^{S} |E(t,s)-\hat{E}(t,s)| $$
>
> In essence, we will sum up the absolute difference between the predicted and ground truth energy arrays for every frame and structure. Afterwards, we take the mean across all frames and structures in the batch. This is the method by which we calculate the global energy MAE. To calculate the per-atom energy MAE, we divide the energy of each structure/frame pair by the number of atoms in the structure. For final frame energy MAE evaluation, we sum up the absolute differences for all structures in the batch only (1 frame) and average it across the number of structures in that batch.

---

> ### Author Response · Authors · 2025-11-23
> **Official Comment by Authors: Part 4**
>
> **Q5. For systems with varying numbers of frames, the authors sample n equidistant points. Does this result in different effective time intervals between samples? If so, could this introduce ambiguity in the learned dynamics or physical consistency of the gradient field?**
>
> **_Response_**: This is a very good point and is one of the limitations we listed at the end of the manuscript. However, we would like to emphasize that the prediction quality is not contingent on the frame counts. Just for general consistency, we sampled n equidistant points from the trajectories, but we would like to point out that the most important parts are the initial and final frames; intermediate frames exist to guide the physics (i.e., provide physical constraints) and trajectory evolution, so these frames can take on different configurations without any issues. In a sense, this also provides a good form of regularization, which helps with generalizability. This is in line what we stated in the manuscript, namely the fact that:
> >… DFT time, in this context, is not real physical time, rather a pseudotime (similar to those in score-based generative and normalizing flow models). Since we are observing the problem from an optimization standpoint, quasi-Newton methods with artificial timesteps should suffice as long as the natural order of the relaxation trajectory is preserved. The learnable network should intrinsically learn the surrogate progression rule (i.e., surrogate dynamic) to equilibrium, since force is still evaluated at each step.
>
> Similarly, the original length of a DFT trajectory does not bear too much weight as long as the initial and final states are defined. During the featurization stage, we have observed trajectories with as little as 3 frames (i.e., long relaxation trajectories not available) and as many as 100 frames. Even with just 3 frames, the model should be able to arrive at the final relaxed state, so the number of frames can be more arbitrary. We noticed this phenomenon in the ablation studies for the explicit latent state evolution case from Table 2 (in the revised manuscript). Upon varying the number of frames for the same unique structure count, we found that RelaxNet was able to produce low energy MAEs for all scenarios, which shows that the frame count sensitivity during training and inferencing is not the sole determining factor, and the model is still very generalizable to the unseen test set structures of diverse composition and conformations.
>
> **General note: poor presentation**
>
> We really appreciate the Reviewer’s assessment of the paper presentation. We reworked the manuscript and would like to summarize the attempts to improve the overall presentation:
> - Added more technical justification/details to key sections
> - Reorganized the general structure of the “Experiments” section with more suitable/descriptive headings and subsections to also accommodate new experiments/results
> - Redrew Figure 3 from the old manuscript (now Figure 3a) to better synthesize all the information and improve aesthetics/clarity
> - Improved the general language and writing quality of entire manuscript
>
> We hope that our responses and revisions to the manuscript sufficiently address the original questions and concerns. To get more comprehensive insights into the contributions of our work, the Reviewer can take a look at our response to Reviewer J9gF’s Q7. Please feel free to raise additional questions, as we are happy to continue the discussion and further improve our work!

---

> > ### Comment · Reviewer_w4Rk · 2025-11-25
> >
> > Thank the authors for the responses!
> >
> > I have two more questions:
> > * For the second concern, the reported MAE range of 2.7-4.9 eV in final-frame energy prediction substantially exceeds the threshold for chemical accuracy (0.043 eV ≈ 1 kcal/mol). Given this magnitude of error, could the authors elaborate on how RelaxNet remains practically useful for this setting?
> > * Regarding Q4, I suggest a more comprehensive assessment. Since RelaxNet might not converge to the ground-truth equilibrium structures, the deviations of its predictions should be evaluated in terms of both geometric accuracy (e.g., via RMSD) and energy accuracy, the latter preferably by performing DFT single-point calculations on both the predicted and true structures for a direct comparison.

---

> > > ### Author Response · Authors · 2025-11-26
> > > **Official Comment by Authors**
> > >
> > > Thank you very much for taking the time to read through all the responses and asking these great questions!
> > >
> > > **Q1. For the second concern, the reported MAE range of 2.7-4.9 eV in final-frame energy prediction substantially exceeds the threshold for chemical accuracy (0.043 eV ≈ 1 kcal/mol). Given this magnitude of error, could the authors elaborate on how RelaxNet remains practically useful for this setting?**
> > >
> > > **_Response_**: Thank you very much for your keen observation. While the global MAE (units: eV) can be a good metric for comparing the overall energy of the system in the case of our ablation study, our focus was on quantifying the model’s overall energy prediction performance as a function of training dataset size and trajectory length with structures that have similar number of atoms. We believe a more chemically-meaningful metric would be the MAE reported in meV/atom, since it is more irrespective of the number of atoms. This is more important for the 5-frame, all-samples case, because the number of atoms across all structures is more diverse. We then compare this metric to the other static baselines to both show RelaxNet prediction competitiveness and chemical accuracy (usually, up to 100 meV/atom is still very acceptable for chemical screening and aiding in relaxation efforts). Given that our reported value is well-below 100 meV/atom (exhibiting DFT-level or near-DFT-level energy prediction accuracy), we believe that RelaxNet can be generally useful for relaxation trajectory prediction.
> > >
> > > **Q2. Regarding Q4, I suggest a more comprehensive assessment. Since RelaxNet might not converge to the ground-truth equilibrium structures, the deviations of its predictions should be evaluated in terms of both geometric accuracy (e.g., via RMSD) and energy accuracy, the latter preferably by performing DFT single-point calculations on both the predicted and true structures for a direct comparison.**
> > >
> > > **_Response_**: Thank you very much for all these great suggestions. We originally reported the pre-Kabsch MAE and post-Kabsch RMSD values of the final frame in the “Benchmarking” section (where we discuss RelaxNet vs. MLIP-based relaxation methods) but moved these metrics to the “Smooth structural relaxation trajectory” subsection under Section 5.4 instead in the newest revisions. We also completely agree with the Reviewer on the energy accuracy quantification and originally considered the single-point DFT calculations as well, but we unfortunately do not have access to VASP, and therefore the original JARVIS DFT workflow (including the correct pseudopotentials, cutoffs, occupation/smearing parameters, SCF convergence settings, etc.), which may ultimately affect true reproducibility. However, we justify that the energy accuracy is present in our predicted final configuration based on the small final-frame force MAE and geometry MAE/RMSD, and the fact that $\textbf{F} = f(\textbf{r},E)$ in our model, so these three physical parameters are inherently linked. By the virial theorem ($T$= kinetic energy, $\textbf{r}$= position, and $\textbf{F}$= forces), which relates the kinetic and potential energies of a system,
> > >
> > > $$2〈T〉+〈\textbf{r} \cdot \textbf{F}〉=0$$
> > >
> > > for an equilibrated structure (i.e., a relaxed structure), small forces (which we demonstrate in our results) indicate that the structure is close to the PES minimum. Additionally, by the Taylor expansion of energy,
> > >
> > > $$E(\hat{\textbf{r}})=E(\textbf{r}_0)+\nabla E(\textbf{r}_0) \cdot (\hat{\textbf{r}}-\textbf{r}_0)+\frac{1}{2} (\hat{\textbf{r}}-\textbf{r}_0)^T H(\hat{\textbf{r}}-\textbf{r}_0)+⋯$$
> > >
> > > when the forces are small ($\nabla E(\textbf{r}_0) \approx 0$), the first-order term completely vanishes and the equations become dominated by the quadratic second-order term, meaning that for small coordinate RMSD and forces, energy errors are also small by extension. Thus, the predicted structures should effectively have a reliable energy estimate even without a DFT single-point calculation.
> > >
> > > We added a few additional lines to the end of “Smooth structural relaxation trajectory” subsection as well to emphasize this point:
> > >
> > > >We also justify energy accuracy in our predicted final configurations based on the small final-frame force and geometry errors, and the inherent $\textbf{F} = f(\textbf{r},E)$ parametric relationship. Thus, by the virial theorem and Taylor expansion of energy, energy deviations are also small by extension.
> > >
> > > We hope these justifications are sufficient, but regardless, we welcome any additional questions, and thank you again for the comments and feedback!

---

> > > > ### Comment · Reviewer_w4Rk · 2025-11-28
> > > >
> > > > Thank the authors for the further responses, which have resolved several of my concerns. Accordingly, I will be raising the rating to 4.

---

> > > > > ### Author Response · Authors · 2025-12-02
> > > > > **Official Comment by Authors**
> > > > >
> > > > > We greatly appreciate the increase in score! We thank you again for the helpful discussion!

---

### Author Response · Authors · 2025-12-02
**Thank you to all Reviewers**

We would like to take the opportunity to thank all the Reviewers who have generously allocated time to thoroughly read and provide valuable input on our manuscript. We truly appreciate all the questions, comments, and feedback that have allowed us to substantially improve our work through added experiments, clearer explanations, and better organization. We would also like to acknowledge Reviewer w4Rk and Reviewer J9gF for taking the time to respond and engage in insightful, continued discussions.

---

### Author Response · Authors · 2025-12-02
**Summary of rebuttal**

We thank the new AC for graciously looking at our work in light of the recent situation. Below is a summary of the rebuttal used to address the Reviewers’ questions and concerns.

**1. New experiments**

- **_Updated Table 1_** to include EGNN benchmark (with training time information), which reveals RelaxNet’s accurate energy predictions (critical for a transient, state-dependent model where errors can accumulate)
- **_Reworked Figure 3_** to better synthesize energy/force plots (Figure 3a); **_added Figure 3b-c_** to better portray geometric pathways and show usefulness of intermediates states, trajectory smoothness, and physical plausibility
- **_Added Table 3_** to emphasize the model’s robustness to perturbations
- **_Added Table 5_** to compare RelaxNet with MLIP-based structure relaxation

**2. Chemically meaningful evaluation metrics**
- **_Energy MAE_** (PES accuracy): conveyed by benchmarking with other state-of-the-art static energy prediction models (Table 4) and MLIP-based structure relaxation (Table 5)
- **_Force MAE_** (PES curvature and relaxation path): we reported low force deviations at the final frame in both our trajectorial ablation study (Table 2) and MLIP-based structure relaxation benchmarking (Table 5)
- **_Geometric MAE/RMSD_** (bond lengths/angles and atomic arrangements): we show smooth, continuous, and physical trajectories via histogram-based per-frame coordinate MAE and geometric pathway plots (Figure 3b-c). We also explicitly report the final coordinates’ MAE/RMSD in the main text.

**3. Model robustness/generalizability**
- We demonstrate robustness to perturbations at both the energy and force level (Table 3)
- We confirm that RelaxNet is not sensitive to trajectory length and frame counts, as shown by trajectorial ablation studies (Table 2)

**4. Reasons for using JARVIS database**
- Most popular datasets (e.g., QM9, MP) do not contain DFT trajectories, only static frames. Trajectory-containing datasets are scarce.
- rMD17 dataset contains MD trajectories for individual small organic molecules (e.g., aspirin, benzene, ethanol), but has very little compositional (and therefore energy magnitude) variation
- JARVIS DFT database contains both composition and conformational diversity, making it better suited for downstream generalizability

**5. Concerns about training time**
- We observe fast convergence despite long per-epoch training time (Table 1)
- We prioritize improvements to energy/force predictions over training time, since the model is transient and heavily state-dependent (errors can propagate over time if intermediate representations are not adequate)
- RelaxNet inferencing is very fast (faster than MLIP-based structure optimization)

**6. Soundness and contributions of the work**
- We provide a very versatile joint neural ODE and graph-based neural network framework for solving dynamical problems. Due to the formulation, RelaxNet improves final state predictions and guarantees physical results because:
    - Neural ODEs learn derivatives, enabling smooth, continuous, and physically consistent trajectories
    - End-to-end training guided by intermediate frames provides internal physical constraints and prevents error accumulation
- Due to RelaxNet high-quality predictions, as shown through benchmarking, it can:
    - Accelerate DFT through better initial guesses
    - Serve as an alternative to MLIP-based relaxation

**7. Advantage of RelaxNet over MLIP-based structure relaxation**
- RelaxNet is a single-step model with end-to-end joint geometry/energy/force training, while MLIP-based relaxation methods consist of two disjointed steps (MLIP training $\rightarrow$ optimization algorithm)
- RelaxNet has better energy/force predictions at the final frame and no unphysical divergences; meanwhile, the quality of the MLIP matters greatly (i.e., if the learned PES is poor, the convergence path towards a relaxed structure will also pose great challenge)
- Several works (e.g., Fu et al. [1]) have found that despite accurately predicting force fields, the reconstitution of dynamic trajectories may not always be as accurate, thus motivating the need for an end-to-end model that can predict these optimization trajectories without instabilities or unphysicalities
- RelaxNet has faster inferencing
- RelaxNet requires less domain knowledge overall, decreasing barrier to entry

**8. Improvements to presentation**
- Reorganized “Experiments” section and renamed subsections with more descriptive headings
- Ensured better aesthetics to the figures
- Improved overall clarity/language quality and included more technical reasonings for key sections
- Clarified variable naming and formulations to ensure consistency

[1] Fu, X., Wu, Z., Wang, W., Xie, T., Keten, S., Gomez-Bombarelli, R. and Jaakkola, T., 2022. Forces are not enough: Benchmark and critical evaluation for machine learning force fields with molecular simulations. (https://openreview.net/forum?id=A8pqQipwkt)

---

### Meta-Review · Area_Chair_xatM · 2026-01-05

**Summary:**

The paper proposes RelaxNet, which combines neural ODEs with message-passing neural networks to model the energy relaxation landscape between initial and final atomic structures. The reviewers agree that the method represents a promising alternative approach to the structure relaxation problem. However, several shared concerns informed the suggested decision. These include the high training cost of the proposed model and the reliance on datasets that contain full relaxation trajectories, which are relatively uncommon and therefore limit the scope of evaluation.

**Reviewer Concerns:**

The authors demonstrate that RelaxNet can produce relaxed coordinates with substantially lower MAE and RMSD compared to structures optimized using MLIP-based approaches, which partially addresses the concern regarding limited and weak baselines. In addition, the sensitivity to perturbations is addressed through additional experiments provided in the rebuttal.

However, other concerns are only partially resolved. The authors clarify that datasets containing relaxation trajectories are scarce and that their experiments are therefore limited to the JARVIS dataset. While this explanation is reasonable, it also reinforces concerns about the general applicability of the method. RelaxNet is designed specifically for the structure relaxation task, whereas static MLIP models can be applied to a broader range of downstream tasks. Combined with the substantially higher training cost of this task-specific model, questions remain regarding its practicality and overall impact.

**Reviewer Scores:**

Reviewer w4Rk has indicated that they would raise their score from 2 to 4.

Reviewer HMBn may raise their score from 4 to 6, in light of the added robustness experiments and the improved relaxation trajectories demonstrated in the rebuttal.

Reviewer J9gF may raise their score from 2 to 4, as the rebuttal provides a clearer and more convincing comparison with static MLIP baselines.

---

### Decision · Program_Chairs · 2026-01-26

Reject